# Sociodemographic data and APOE-ε4 augmentation for MRI-based detection of amnestic mild cognitive impairment using deep learning systems

Obioma Pelka[1,2], Christoph M. Friedrich[1,4]*, Felix Nensa[2], Christoph Mönninghoff[3], Louise Bloch[1,4], Karl-Heinz Jöckel[4], Sara Schramm[4], Sarah Sanchez Hoffmann[5], Angela Winkler[5], Christian Weimar[5☯], Martha Jokisch[5☯], for the Alzheimer's Disease Neuroimaging Initiative[¶]

**1** Department of Computer Science, University of Applied Sciences and Arts Dortmund (FHDO), Dortmund, NRW, Germany, **2** Department of Diagnostic and Interventional Radiology and Neuroradiology, University Hospital Essen, University of Duisburg-Essen, Essen, NRW, Germany, **3** Clinic for Neuroradiology, Clemens Hospital, Münster, NRW, Germany, **4** Institute for Medical Informatics, Biometry and Epidemiology (IMIBE), University Hospital Essen, University of Duisburg-Essen, Essen, NRW, Germany, **5** Department of Neurology, University Hospital of Essen, University of Duisburg-Essen, Essen, NRW, Germany

☯ These authors contributed equally to this work.
¶ Membership of the Alzheimer's Disease Neuroimaging Initiative is listed in the Acknowledgments.
* christoph.friedrich@fh-dortmund.de

**Data Availability Statement:** De-identified data from this study are available upon request to Department of Neurology, University Hospital

## Abstract

Detection and diagnosis of early and subclinical stages of Alzheimer's Disease (AD) play an essential role in the implementation of intervention and prevention strategies. Neuroimaging techniques predominantly provide insight into anatomic structure changes associated with AD. Deep learning methods have been extensively applied towards creating and evaluating models capable of differentiating between cognitively unimpaired, patients with Mild Cognitive Impairment (MCI) and AD dementia. Several published approaches apply information fusion techniques, providing ways of combining several input sources in the medical domain, which contributes to knowledge of broader and enriched quality. The aim of this paper is to fuse sociodemographic data such as age, marital status, education and gender, and genetic data (presence of an apolipoprotein E (APOE)-ε4 allele) with Magnetic Resonance Imaging (MRI) scans. This enables enriched multi-modal features, that adequately represent the MRI scan visually and is adopted for creating and modeling classification systems capable of detecting amnestic MCI (aMCI). To fully utilize the potential of deep convolutional neural networks, two extra color layers denoting contrast intensified and blurred image adaptations are virtually augmented to each MRI scan, completing the Red-Green-Blue (RGB) color channels. Deep convolutional activation features (DeCAF) are extracted from the average pooling layer of the deep learning system Inception_v3. These features from the fused MRI scans are used as visual representation for the Long Short-Term Memory (LSTM) based Recurrent Neural Network (RNN) classification model. The proposed approach is evaluated on a sub-study containing 120 participants (aMCI = 61 and cognitively unimpaired = 59) of the Heinz Nixdorf Recall (HNR) Study with a baseline model

Essen, Germany pending approval by the study steering committee. Due to data security reasons (i.e., data contain potentially participant identifying information), the HNR Study does not allow sharing data as public use file. Data requests can be addressed to: recall@uk-essen.de Additional Data used in preparation of this article were obtained from the Alzheimer's Disease Neuroimaging Initiative (ADNI) database (https://adni.loni.usc.edu). Details about data access is detailed there. The authors had no special access privileges others would not have to the data obtained from the Alzheimer's Disease Neuroimaging Initiative (ADNI) database.

**Funding:** The work of Obioma Pelka was partially funded by a PhD grant from University of Applied Sciences and Arts Dortmund, Germany. The authors thank the Heinz Nixdorf Foundation [Chairman: Martin Nixdorf; Past Chairman: Dr jur. Gerhard Schmidt (†)], for their generous support of this study. Parts of the study were also supported by the German Research Council (DFG) [DFG project: EI 969/2-3, ER 155/6-1;6-2, HO 3314/2-1;2-2;2-3;4-3, INST 58219/32-1, JO 170/8-1, KN 885/3-1, PE 2309/2-1, SI 236/8-1;9-1;10-1,], the German Ministry of Education and Science [BMBF project: 01EG0401, 01GI0856, 01GI0860, 01GS0820_WB2-C, 01ER1001D, 01GI0205], the Ministry of Innovation, Science, Research and Technology, North Rhine-Westphalia (MIWFT-NRW), the Else Kröner-Fresenius-Stiftung [project: 2015_A119] and the German Social Accident Insurance [DGUV project: FF-FP295]. Furthermore the study was supported by the Competence Network for HIV/AIDS, the deanship of the university hospital and IFORES of the university Duisburg-Essen, the European Union, the German Competence Network Heart Failure, Kulturstiftung Essen, the Protein Research Unit within Europe (PURE) and the following companies: Celgene GmbH München, Imatron/GE-Imatron, Janssen, Merck KG, Philips, ResMed Foundation, Roche Diagnostics, Sarstedt AG&Co, Siemens HealthCare Diagnostics, Volkswagen Foundation. This substudy conducted in the Department of Neurology of the university hospital Essen was additionally supported by the Dr. Werner-Jackstädt Stiftung and Janssen.

**Competing interests:** The authors have declared that no competing interests exist.

accuracy of 76%. Further evaluation was conducted on the ADNI Phase 1 dataset with 624 participants (aMCI = 397 and cognitively unimpaired = 227) with a baseline model accuracy of 66.27%. Experimental results show that the proposed approach achieves 90% accuracy and 0.90 $F_1$-Score at classification of aMCI vs. cognitively unimpaired participants on the HNR Study dataset, and 77% accuracy and 0.83 $F_1$-Score on the ADNI dataset.

## Introduction

Alzheimer's disease (AD) is a progressive neurodegenerative disease that causes behavioral changes and deterioration of memory and other cognitive domains [1]. Because there is no causal treatment for AD dementia, identifying early stages of the disease and preclinical markers will help to implement intervention and prevention strategies [2]. Mild cognitive impairment (MCI) is a clinical entity that describes the stage between cognitive changes of normal aging and dementia [3, 4]. The amnestic MCI (aMCI) subtype has a high probability of progressing to AD dementia [2]. Robust and reliable systems for early aMCI classification that aid doctors to identify high risk individuals are needed.

Individuals with aMCI have a higher risk to develop AD dementia, but some individuals also revert to normal or stay stable without reaching the AD dementia stage [5]. Thus, it would be beneficial to implement an additional classification system for progression that does not need any invasive biomarker assessments (like beta-amyloid or tau in the cerebrospinal fluid). Magnetic resonance imaging (MRI) techniques offer a broad visual representation, that can be adopted for this purpose. For an effective classification of images, the selection and combination of adequate features, and labeled training data is crucial. The more knowledge present, the more enriched image representations are available. The selection and combination of features for an adequate representation of the images is essential for creating effective classification systems. Several research explorations using multi-modal representations and aiming to sufficiently represent biomedical and medical images achieve higher prediction accuracies. In Codella et al. [6], automated medical image modality recognition was achieved by fusing visual and text information. Valavanis et al. [7] and Pelka et al. [8] adopted a combination of visual representation with text information extracted from captions to classify and predict the image modality at the ImageCLEF2016 Medical Task [9].

Deep learning techniques [10] have improved prediction accuracies in object detection [11], speech recognition [12] and in medical imaging [13, 14]. These positive results are attributable to large amounts of natural scene data sets available, as they provide adequate feature representation for transfer learning [15]. However, a major concern in the medical domain is the insufficient number of large datasets such as ChestX-Ray8 database [16] and the Open-fMRI project [17]. This is due to the fact that detailed annotation of medical images is time-consuming, prone to errors and restricted by data protection rules. Therefore, image classification tasks in the medical domain are challenging, regarding sufficient and efficient feature selection. On the other hand, there are several input sources in the medical domain. These can be fused together, such as combining MRI with patient clinical information or several imaging modalities, as well as radiology reports with images, to obtain better medical image understanding. There is no restriction to the usage of the fused data, as it can be applied to several challenging medical tasks.

## Related work

Successful research work regarding the prediction of the conversion from mild cognitive impairment to Alzheimer's disease have been reported using multimodal features from several input sources. Spasov *et al.* [18] proposed a MCI to AD conversion and AD vs. healthy controls detection using deep learning techniques to combine structural MRIs with demographic, neuropsychological and APOE-$\varepsilon 4$. The proposed model is based on dual learning and an **ad hoc** layer for 3D separable convolutions. Generative methods that detect occuring patterns were applied by Yang *et al.* [19] to characterize Alzheimer's Disease using image and categorical genetic features, based on supervised topic modeling. In Lee *et al.* [20], a multimodal recurrent neural network using demographic information, longitudinal cognitive performance and cross-sectional neuroimaging biomarker was adopted for MCI to AD conversion prediction. The experimented objective was a sequential data classification and several Gated Recurrent Unit (GRU) for each data modality were trained and adopted for MCI prediction.

Several prior works Zhang *et al.* [21], Liu *et al.* [22], Samper-González *et al.* [23] and Huang *et al.* [24] apply machine learning and neuroimaging to distinguish between cognitively unimpaired controls and patients with MCI and AD. A traditional way is to first extract features like volume, cortical thickness or gray matter volume from neuroimaging and then perform feature selection, as well as dimension and noise reduction. Finally, a feature-based classification is then conducted. This approach has been presented in multiple research work including Bloch *et al.* [25] and Sørensen *et al.* [26]. Choosing the best feature combination for several medical tasks can be tedious, time-consuming and challenging. As automatic feature-extraction from 3D-images is often combined with high computational effort, Liu *et al.* [22] and Huang *et al.* [24] use deep learning methods to extract information directly from the MRI scans, which improves the overall classification results.

Multimodal approaches have shown to obtain encouraging results in other domains as well, such as biomedical image analysis. These attempts combine image and text representation into one vector, with which the image classifiers are trained. Adopting this method, the connections in low-level features can be exploited. For the ImageCLEF 2015 Medical Tasks [27], late fusion methods were applied in Pelka *et al.* [28] to fuse decision values from a multiclass linear kernel Support Vector Machine (SVM) [29] and Random Forest [30] classifiers to predict the modality of subfigures extracted from the PubMed Central (PMC) Open Access Subset [31]. In [32], automatic generated semantic information from Unified Modeling Language System (UMLS) [33] concepts were combined with Bag-of-Keypoints representations [34] computed with Dense (dSIFT) [35] features and applied for predicting image modality, body region examined, orientation of the image and biological system investigated. This approach was further explored in Pelka *et al.* [36] by using Deep convolutional activation features (DeCAF) [37] to obtain an optimized medical image body region classification.

Inspired by this and earlier work on body region detection in Pelka *et al.* [38], we propose an approach that brands encoded sociodemographic and genetic data onto MRI 2D slices to obtain an enhanced image representation, to reduce computational load.

Due to the limited number of annotated medical images available, we propose to learn augmented deep convolutional activation features in a recurrent neural network framework for an optimized aMCI classification. These features are extracted with the Inception_v3 [39] deep learning model, thereby exploring the potential of Transfer Learning [15] from pre-trained ImageNet models. Promising results using deep convolutional activation features (DeCAF) have been presented by various work, including Gong *et al.* [40], Yosinski *et al.* [41], Sinha *et al.* [42] and Razavian *et al.* [43]. Our contributions in this paper are:

- A novel fusion method for branding MRI scans with patient sociodemographic and genetic data.

- Enhancement of MRI scans by augmenting two extra color layers.

- Transfer Learning is utilized for creating deep convolutional activation features.

- Long short-term memory (LSTM) based Recurrent Neural Networks (RNN) are utilized for modeling approaches

- Evaluation on sub sample of the Heinz Nixdorf Recall (HNR, Risk Factors, Evaluation of Coronary Calcium and Lifestyle) Study with 1.5T-weighted MRI scans.

- Further evaluation was conducted on the Alzheimer's Disease Neuroimaging Initiative (ADNI) Phase 1 dataset with 1.5T-weighted MRI scans.

## Materials and methods

### Study population

The proposed data fusion techniques were evaluated using a sub sample of 61 participants with aMCI and 59 cognitively unimpaired controls derived from the Heinz Nixdorf Recall (HNR) Study [44] and further evaluated on the ADNI Phase 1 dataset, an open-accessible state-of-the-art ADNI Phase 1 dataset distributed by the Alzheimer's Disease Neuroimaging Initiative (https://adni.loni.usc.edu) [45].

   The HNR Study is a population-based prospective cohort study with subjects randomly selected from mandatory lists of residence. Its major aim is to evaluate the predictive value of coronary artery calcification using electron-beam computed tomography for myocardial infarction and cardiac death in comparison to other cardiovascular risk factors. Details of the study methods have been previously described in detail [44]. Ethics Statement for the use of the HNR study population from IRB of University Hospital Essen, Essen, Germany dated 2009-10-23 and 2012-06-06 to Prof. Dr. C. Weimar, registration number: 06-3116 is available and was approved by the university review board. All participants provided written informed consent.

   Briefly, 4814 participants 45 to 75 years of age were enrolled between 2000 and 2003 in the Ruhr area in Germany. Five years after baseline (2005-2008, n = 4,145), the first follow-up of the HNR Study was conducted and included a short cognitive assessment (for details see [46]). This cognitive assessment was evaluated and validated in a sub-study [46]. The longitudinal sub-study comprises a more comprehensive neuropsychological assessment (see below), a neurological exam assessed by a certified neurologist and MRI volumetric data [1, 46]. Participants with dementia (n = 7), severe depression (ADAS depression subscale score >4, n = 13), Parkinson disease (n = 5), mental retardation (n = 2), severe alcohol consumption (for women: >20 g/day; for men: >40 g/day, n = 2), known brain cancer (n = 1), severe problems with the German language (foreign persons, n = 9) and severe sensory impairment (n = 2) leading to invalid cognitive testing were excluded from the sub-study.

   ADNI is a consortium of several medical centers and universities in the United States and Canada, and was established to create biomarker procedures and standardized imaging techniques in subjects with MCI, subjects with AD, and normal subjects [45]. Led by Principal Investigator Michael W. Weiner, MD., ADNI was launched in 2003 as a public-private partnership. One of the major aims of this initiative was to develop an accessible data repository that contains serial magnetic resonance imaging (MRI), positron emission tomography (PET), other biological markers, and clinical and neuropsychological assessment. Using this

repository, modeling approaches capable off measuring the progression of mild cognitive impairment (MCI) and early Alzheimer's disease (AD) can implemented and evaluated. For up-to-date information, see http://www.adni-info.org and details about the ethics statement of the ADNI study population can be found at https://adni.loni.usc.edu.

All enrolled subjects in the ADNI Phase 1 dataset were between 55 and 90 years of age, could either speak Spanish or English and were classified as normal controls, subjects with MCI or subjects with mild AD [45]. Participants with no memory complaints were classified as normal subjects. The Clinical Dementia Rating (CDR) for normal, MCI, and AD subjects were 0, 0.5 and > 0.5, respectively [45]. All classified subjects had a study partner with over 10 hours contact per week, adequate visual and auditory perceptiveness, at least 6 years of education or similar work biography, and general good health, a geriatric depression score of >= 4 [45]. Female participants had to be either 2 years past child bearing potential or sterile. Further information on subject selection is detailed in Petersen *et al.* [45].

## Evaluation of cognitive status and aMCI diagnosis

Detailed information on the neuropsychological assessment to identify participants with MCI of the HNR Study has been described in Dlugal *et al.* [47]. Briefly, the standardized neuropsychological examination was conducted by a neuropsychologist using the following tests:

1. The Alzheimer's Disease Assessment Scale (ADAS) [48]

2. Number Connection Test from the NAI [49]

3. Verbal Fluency Test [50] (two subtests with a formal lexical category and two subtests with a semantic category)

4. Instrumental Activities of Daily Living scale to assess disability [49]

Using these tests, the following areas of neuropsychological functioning were covered: verbal memory, orientation/praxis, information processing speed, executive functions and verbal abilities. A cognitive domain was rated as impaired if the performance was more than 1 standard deviation (SD) below the age adjusted mean.

Because the MCI due to AD criteria by Albert et al. [51] were not yet published when the sub-study started, the Winblad et al. [52] MCI criteria were used to diagnose aMCI. The 61 aMCI participants had to meet all of the following aMCI criteria:

1. cognitive impairment in the verbal memory domain

2. subjective cognitive decline

3. normal functional abilities and daily activities

4. no dementia diagnosis

The final decision about aMCI diagnosis was ultimately made by consensus agreement between the examining neurologist and neuropsychologist taking into account the medical history related to cognitive functioning, duration of such symptoms, the history of other medical illnesses and current treatment for each participant. The diagnosis aMCI is equivalent to the diagnosis of MCI due to AD without biomarker information representing the core clinical criteria as proposed by Albert et al. [51]. Participants who did not show cognitive impairment in any domain were considered as cognitively unimpaired and categorized as "Controls".

For the ADNI Phase 1 Population, the Assessment is detailed in Petersen *et al.* [45].

## Covariates

To fuse several input sources in the medical domain, MRI combined with the following socio-demographic characteristics were used: age, gender, education and marital status. Education was classified by the International Standard Classification of education (ISCED) as total years of formal education, combing school and vocational training [53]. For the HNR Study, the continuous education variable was grouped into three categories, with the highest category of 14 and more years of education and the lowest category with 10 and fewer years. Participants were asked about their marital status using the following categories (married, widowed, divorced and single). For the ADNI Phase 1 dataset, education was based on the Logical Memory II subscale of the Wechsler Memory Scale-Revised [54] and on subject classification. For normal subjects, the cutoff scores were $>= 9$ for 16 years of education, $>= 5$ for 8 to 15 years of education and $>= 3$ for 0 to 7 years. For subjects with MCI and subjects with AD, the cutoff scores were $<= 8$ for 16 years of education, $<= 4$ for 8 to 15 years of education and $<= 2$ for 0 to 7 years.

Furthermore, genetic information was adopted for the proposed fusion approach prior to training the classification model. The apolipoprotein E (APOE)-$\varepsilon 4$ allele is the main genetic risk factor for sporadic AD [55]. For the HNR Study, Cardio-MetaboChip BeadArrays were used for genotyping of two single-nucleotide polymorphisms (rs7412 and rs429358) to discriminate between the APOE alleles $\varepsilon 2$, $\varepsilon 3$, and $\varepsilon 4$. Participants defined as APOE-$\varepsilon 4$ positive had at least one allele 4 (2/4, 3/4, 4/4). All other participants were defined as APOE-$\varepsilon 4$ negative. Information regarding APOE-$\varepsilon$ on the ADNI Phase 1 dataset is detailed in Petersen *et al.* [45].

## Dataset

Table 1 shows the distribution of the sociodemographic data variables age, gender, education, marital status and genetic data variable APOE-$\varepsilon 4$ genotype (defined as "Participant Data") for aMCI and cognitively unimpaired controls on the applied sub sample. All participants were scanned with a single 1.5T MR scanner (Magneton Avanto, Siemens Healthcare, Erlangen) with 60cm bore diameter, 200T/m/s slew rate, 160cm length and 40/40/45 mT/m gradient strength [1].

To additionally examine the proposed approach, the open-accessible state-of-the-art ADNI Phase 1 dataset was used, which is distributed by the Alzheimer's Disease Neuroimaging Initiative (https://adni.loni.usc.edu) [45]. This initiative is a consortium of several medical centers and universities in the United States and Canada, and was established to create biomarker procedures and standardized imaging techniques in subjects with MCI, subjects with AD, and normal subjects [45]. Led by Principal Investigator Michael W. Weiner, MD., ADNI was launched in 2003 as a public-private partnership. One of the major aims of this initiative was to develop an accessible data repository that contains serial magnetic resonance imaging (MRI), positron emission tomography (PET), other biological markers, and clinical and neuropsychological assessment. Using this repository, modeling approaches capable off measuring the progression of mild cognitive impairment (MCI) and early Alzheimer's disease (AD) can be implemented and evaluated. For up-to-date information, see http://www.adni-info.org. Table 2 shows the distribution of the participant data variables used for this evaluation from the ADNI Phase 1 dataset.

## Data fusion

The presented work proposes an approach to fuse sociodemographic data and APOE-$\varepsilon 4$ with MRI scans, enabling enriched multi-modal image representation. This is fundamental for

**Table 1. HNR Study explorative analysis.** Summary statistics computed on the sub-study of the HNR Study adopted for the proposed fusion approach. Participant Data denotes the sociodemographic data (age, marital status, education, gender) and genetic data (APOE-$\varepsilon$4). The total number of particpants is $n = 120$.

| Participant Data | | aMCI | Controls | Sum |
|---|---|---|---|---|
| age$_{yr}$ | 46–55 | 1 (50.00%) | 1 (50.00%) | 2 (1.67%) |
| | 56–65 | 15 (60.00%) | 10 (40.00%) | 25 (20.83%) |
| | 66–75 | 31 (48.44%) | 33 (51.56%) | 64 (53.33%) |
| | 76–85 | 14 (48.28%) | 15 (51.72%) | 29 (21.17%) |
| gender | Female | 24 (50.00%) | 24 (50.00%) | 48 (40.00%) |
| | Male | 37 (51.39%) | 35 (48.61%) | 72 (60.00%) |
| education$_{yr}$ | <= 10 | 15 (55.56%) | 12 (44.44%) | 27 (22.50%) |
| | 11–13 | 37 (53.62%) | 32 (46.38%) | 69 (57.50%) |
| | >= 14 | 9 (37.50%) | 15 (62.50%) | 24 (20.00%) |
| marital status | Married | 49 (51.04%) | 47 (48.96%) | 96 (80.00%) |
| | Widowed | 8 (57.14%) | 7 (42.86%) | 14 (11.67%) |
| | Divorced | 4 (57.14%) | 3 (42.86%) | 7 (5.83%) |
| | Single | 0 (0%) | 2 (100%) | 2 (1.67%) |
| APOE-$\varepsilon$4 | Positive | 21 (63.64%) | 12 (36.36%) | 33 (27.50%) |
| | Negative | 40 (45.98%) | 47 (54.02%) | 87 (72.50%) |
| Sum | | 61 (50.83%) | 59 (49.17%) | 120 (100.00%) |

aMCI = Amnestic Mild cognitive impairment

Controls = Cognitively unimpaired

**Table 2. ADNI Phase 1 dataset explorative analysis.** Summary statistics computed on ADNI Phase 1 dataset adopted for the proposed fusion approach. Participant Data denotes the sociodemographic data (age, marital status, education, gender) and genetic data (APOE-$\varepsilon$4). The total number of particpants is $n = 624$.

| Participant Data | | aMCI | Controls | Sum |
|---|---|---|---|---|
| age$_{yr}$ | 46–55 | 3 (100.00%) | 0 (00.00%) | 2 (00.48%) |
| | 56–65 | 52 (89.66%) | 6 (10.34%) | 58 (09.29%) |
| | 66–75 | 158 (58.52%) | 112 (41.48%) | 270 (43.27%) |
| | 76–90 | 184 (62.80%) | 109 (37.20%) | 293 (46.96%) |
| gender | Female | 141 (56.40%) | 109 (43.60%) | 250 (40.01%) |
| | Male | 256 (68.45%) | 118 (31.55%) | 374 (59.94%) |
| education$_{yr}$ | <= 10 | 20 (66.67%) | 10 (33.33%) | 30 (04.80%) |
| | 11–13 | 79 (72.48%) | 30 (27.52%) | 109 (17.46%) |
| | >= 14 | 298 (61.44%) | 187 (38.56%) | 485 (77.72%) |
| marital status | Married | 318 (67.23%) | 155 (32.77%) | 473 (75.80%) |
| | Widowed | 48 (55.17%) | 39 (44.83%) | 87 (13.94%) |
| | Divorced | 25 (59.52%) | 17 (40.48%) | 42 (06.73%) |
| | Single | 6 (27.27%) | 16 (72.72%) | 22 (03.53%) |
| APOE-$\varepsilon$4 | Positive | 185 (52.56%) | 167 (47.44%) | 352 (56.41%) |
| | Negative | 212 (77.94%) | 60 (22.06%) | 272 (43.59%) |
| Sum | | 397 (63.62%) | 227 (36.38%) | 624 (100.00%) |

aMCI = Amnestic Mild cognitive impairment

Controls = Cognitively unimpaired

| Participant Data | Values | | | |
|---|---|---|---|---|
| age$_{yr}$ | 46 - 55 ☐ | 56 - 65 ◨ | 66 - 75 ▬ | 76 - 85 ■ |
| gender | Male ■ | Female ☐ | | |
| education$_{yr}$ | <= 10 ■ | 11 - 14 ◨ | >= 14 ◫ | |
| marital status | Married ■ | Widow ◫ | Divorced ▬ | Single ☐ |
| APOE-$\varepsilon$4 | Positive ■ | Negative ☐ | | |

■ 1st branding marker option

◧ 2nd branding marker option

▬ 3rd branding marker option

☐ 4th branding marker option

**Fig 1. Marker for branding.** Generated markers applied for fusing sociodemographic data and APOE-$\varepsilon$4 data with 2D slices of MRI scans. Each marker denotes the different values for clinical data variables. Participant Data denote the sociodemographic data variables (age, marital status, education, gender) and genetic data variable (APOE-$\varepsilon$4). The markers were randomly distributed amongst values per variable.

image classification and retrieval purposes, and is not limited to computer-aided decision systems for clinical diagnoses. Positive results have been presented for 2D Images in Pelka *et al.* [38], where automatic generated keywords were incorporated onto x-ray and biomedical images. Several branding options such numerical, grayscale and ordinal values were experimented in Pelka *et al.*, and the binary branding option proved to obtain best results [38]. This approach is further investigated in this work by encoding sociodemographic data and APOE-$\varepsilon$4 onto 2D slices of an MRI scan for a specific clinical question, and is denoted as "Branded". The limitation of the usage of 2D slices instead of the 3D MRI scans will be experimented in further work, as positive results have been reported in the overview survey of deep learning techniques for MRI [56].

Fusing information from multiple input domain, aims at increasing consolidated representations of the participants. For each of the variables that are listed in sociodemographic data and APOE-$\varepsilon$4, possible values are grouped. Hence, 2 to 4 groups were obtained per variable. To incorporate these groups onto the MRI scans, generated markers displayed in Fig 1 are created as a variable group.

Finally, each 2D slice (image size [224x224]) is branded by markers denoting the participant data values, which are listed in Fig 1. Each participant's information is fused as a [10x20] pixel marker at the pixel position (0, 10) to (10, 150). A space [10x5] is kept between each marker position, as shown in Fig 2 and the complete implementation was done in python and will be

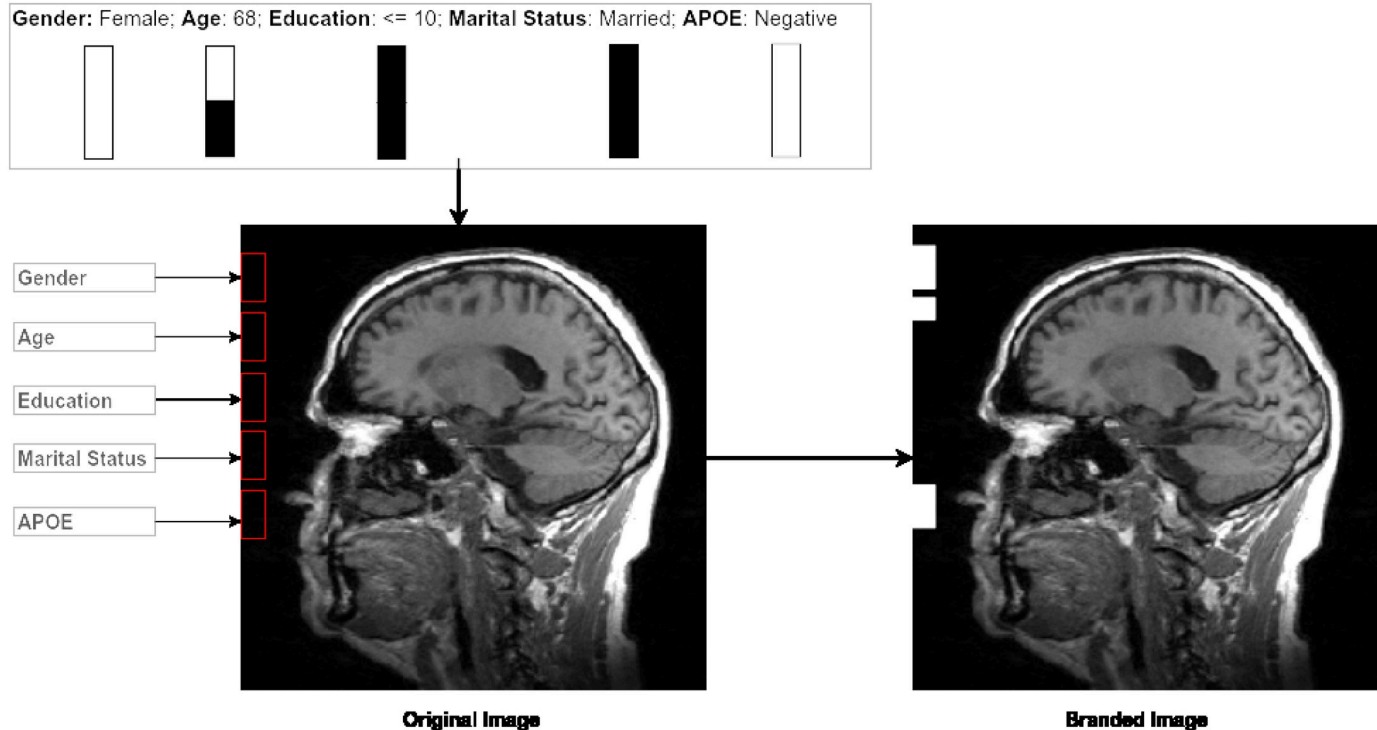

**Fig 2. Branding approach.** Proposed branding approach of fusing sociodemographic data (age, education, marital status and gender) and genetic data (APOE-$\varepsilon$4) with 2D slices of an MRI scan. The marker positions and sizes of each clinical data variable branded are displayed. The 2D slice was randomly selected from an MRI scan of the sub-study from the HNR Study.

available after acceptance. For the HNR Study dataset, all DICOM scans were converted to png-files and resized to [224x224], prior to branding and image enhancement. Similarly, for the ADNI Phase 1 dataset, the nifti scans were converted to png-files. The png-files from both datasets are 8-bit.

## Image enhancement

For image recognition tasks, convolutional neural networks trained on large datasets produce favorable results. Considering the number of images in the applied data set, the adaptation of Transfer Learning with pre-trained neural network Inception-v3 [39] was chosen. This pre-trained deep convolutional neural network models were designed to extract among other features, color information in three separate channels (RGB) from the images [57, 58]. However, the MRI scan are gray-scale and have a single color channel with values 0,. . .,255. To fully utilize the capabilities of deep convolutional neural networks, two extra color layers are augmented to each MRI, completing the Red-Green-Blue (RGB) channels. Color input enhancement have aided to substantially improve prediction accuracy from 86% to 92% for the detection of malignancy in digital mammography images [59] and approximately 3% for structuring 2D x-rays according to imaging technique modality, anatomical region and biological systems examined, which is applied for medical image retrieval [60].

The first extra layer was obtained by using the image processing technique: Contrast Limited Adaptive Historization Equation (CLAHE) [61]. CLAHE is a contrast enhancement method, modified from the Adaptive Histogram Equation (AHE). It is designed to be broadly

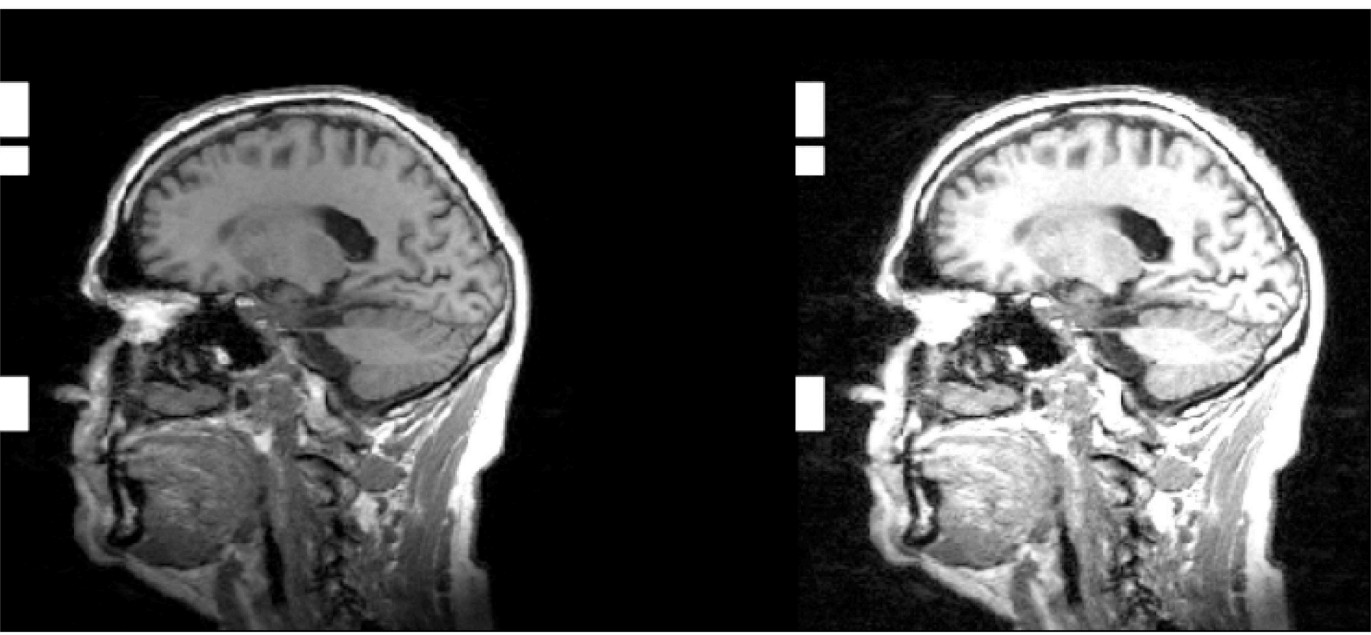

**Fig 3. CLAHE image preprocessing.** 2D slice from a MRI scan before and after applying the Contrast Limited Adaptive Histogram Equation (CLAHE) preprocessing method. The 2D slice was randomly selected from an MRI scan of the sub-study from the HNR Study.

applicable and has demonstrated effectiveness, especially for medical images [62]. Fig 3 displays the original 2D slice of a MRI scan with contrast enhanced image adaption after CLAHE was performed. The CLAHE output image was obtained using the following parameters:

- Desired histogram shape: Uniform

- Distribution parameter: 0.4

- Number of histogram bins: 256

- Contrast enhancement limit: 0.01

- Range of output data: Full

- Number of tiles: [8, 8]

The second layer was generated by applying the Non-Local Means (NL-MEANS) preprocessing method. This is a digital image denoising method, based on a non-local averaging of all present pixels in an image [63]. The effect of applying NL-MEANS to a randomly chosen 2D slice is shown in Fig 4.

The NL-MEANS output images were obtained using the following parameters:

- Filter strength: 0.05

- Kernel ratio: 4

- Window ratio: 4

### Visual representation

For visual representation, deep convolutional activation features (DeCAF) [37] were chosen. DeCAF features are extracted from the average pooling layer of the deep learning system

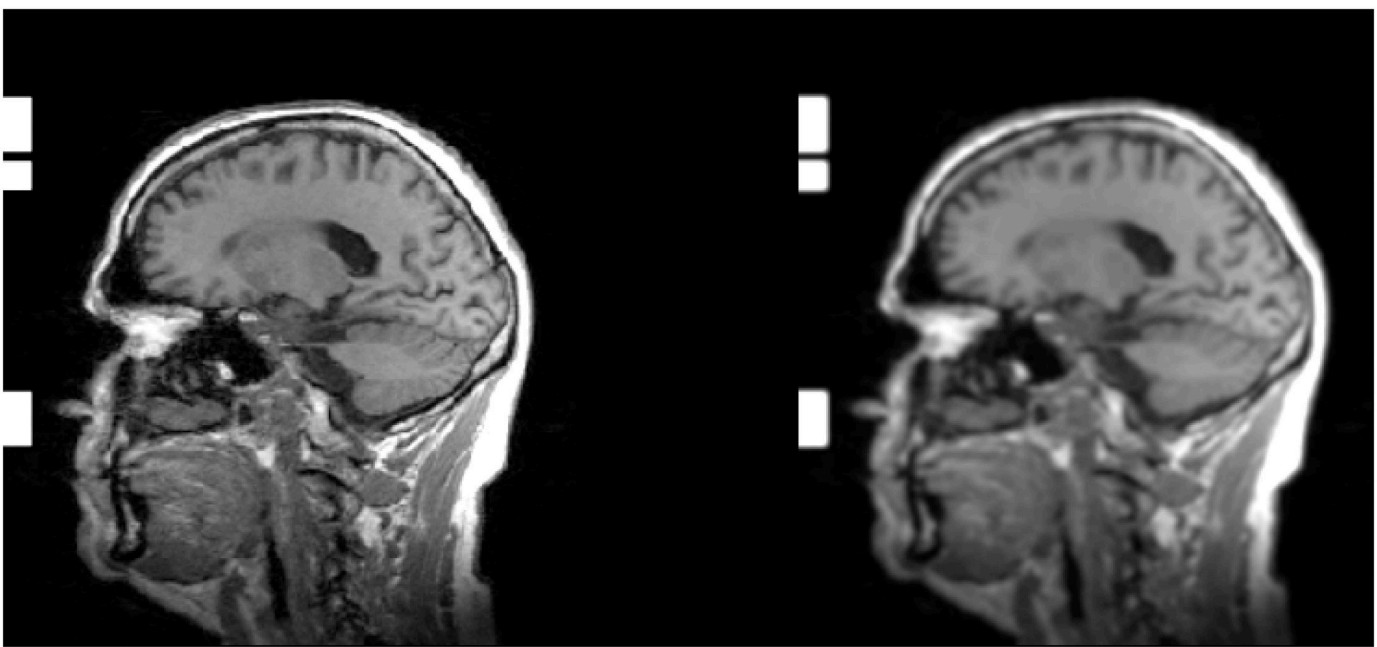

**Fig 4. NL-MEANS image preprocessing.** 2D slice from a MRI scan before and after applying the Non-Local Means (NL-MEANS) preprocessing method. The 2D slice was randomly selected from an MRI scan of the sub-study from the HNR Study.

Inception-v3 [39], which is pre-trained on the ImageNet [64]. For comparison purposes, additional DeCAF features were extracted using a medical context pre-trained DenseNet-121 model [65] on the ChestX-Ray8 database [16]. The activation features were extracted using the neural network API Keras 2.2.0 [66]. The default values for the Inception-v3 base model was used. For the 3D MRI scans, the DeCAF visual representations were extracted 2D slice-wise with a vector size of 2048. Every second 2D slice between [8 − 165] was considered. Hence, each 3D MRI scan was represented with 80 2D slices and has a vector size of 163, 840 deep convolutional activation features.

## Classification

As aMCI vs control classification model, LSTM based RNNs was adopted. RNNs are mostly used for modeling long-range dependencies, where future events are predicted with past events [67] and has proven to be successful for several research topics such as medical question and answering [68]. The effective characteristic of LSTM is the ability to accumulate state information, as information of every new input is accumulated onto previous input [69, 70].

As each 2D slice of a MRI scan contains dependencies between predecessor and successor slices, we choose the LSTM architecture for modeling the classifier. The applied LSTM network contains the following keras layers:

- LSTM

  - Output shape: (None, 2048)

  - Input shape: (80, 2048)

  - Dropout = 0.5

- Dense

- Output shape: (None, 512)

- Activation: Sigmoid $[1/(1 + exp(-x))]$

- Dropout

  - Output shape: (None, 512)

  - Rate: 0.5

- Dense

  - Output shape: (None, 2)

  - Activation: Softmax

For the approach evaluation, three (3) different inputs were fed into the LSTM network:

1. **Original**: DeCAF representations extracted from the original MRI scans.

2. **Branded**: DeCAF representations extracted with the branded and enhanced MRI scans.

3. **Wide and Deep** [71]: Dot product of features extracted using the original MRI and clinical data.

The HNR Study dataset consisting of 120 participants was split into a training and test set, containing 99 participants (aMCI = 51 and controls = 48) and 21 participants (aMCI = 10 and controls = 11), respectively. Similarly, the ADNI Phase 1 dataset with 624 participants was split into a training and test with 561 (aMCI = 357 and controls = 204) and 63 (aMCI = 40 and controls = 23), respectively. The test set was independent and not used for training or parameter optimization. The complete workflow describing the proposed method is displayed in Fig 5.

## Results

For the HNR Study datatset, a $k$ = 5-fold cross validation [72] was achieved by splitting the training set with 99 participants into 5 different partitions. From this, one partition is used as the validation set (20%) and the remaining 4 partitions (80%) are used for training. This has

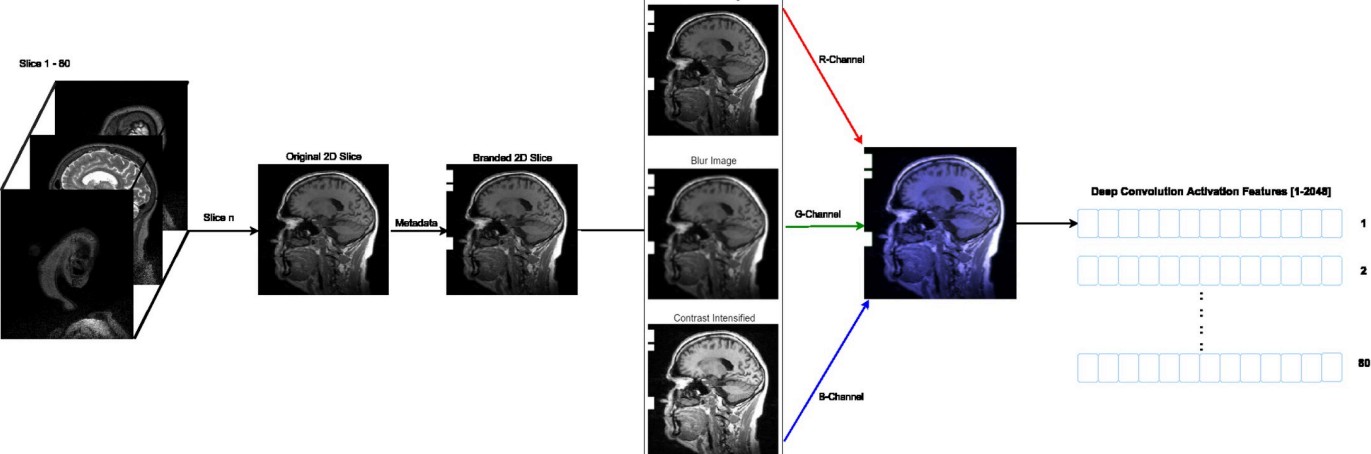

**Fig 5. Complete proposed approach.** Complete workflow of the proposed approach. Sociodemographic data and APOE-$\varepsilon$4 are fused with MRI scans 2D slice-wise and further enhanced by augmenting contrast intensified and blurred image adaptions as two extra layer completing the RGB channels. DeCAF representations are extracted and used as visual representations for training the aMCI vs control classification model.

**Table 3. Cross-validation prediction on HNR Study.** Prediction performance of the LSTM classification model using various image input types. The highlighted values are the best per evaluation metric. Evaluation was calculated on the $k$ = 5-fold cross validation sets from the training set with $n$ = 99 participants of the sub-study from the HNR Study. The values are the average and standard deviation rates across all $k$ = 5 cross validation sets. Visual representation were extracted using the ImageNet database [64].

|  | Original | Branded | Wide and Deep |
|---|---|---|---|
| Specificity | 0.64 (± 0.26) | **0.80 (± 0.18)** | 0.64 (± 0.21) |
| Sensitivity | 0.70 (± 0.12) | 0.74 (± 0.11) | **0.76 (± 0.18)**) |
| $F_1$-Score | 0.69 (± 0.09) | **0.80 (± 0.14)** | 0.71 (± 0.08)) |
| Accuracy | 0.70 (± 0.16) | **0.77 (± 0.07)** | 0.70 (± 0.07) |

**Table 4. Cross-validation prediction on HNR Study.** Prediction performance of the LSTM classification model using various image input types. The highlighted values are the best per evaluation metric. Evaluation was calculated on the $k$ = 5-fold cross validation sets from the training set with $n$ = 99 participants of the sub-study from the HNR Study. The values are the average and standard deviation rates across all $k$ = 5-fold cross validation sets. Visual representation were extracted using the ChestX-Ray8 database [16].

|  | Original | Branded | Wide and Deep |
|---|---|---|---|
| Specificity | 0.68 (± 0.13) | **0.76 (± 0.11)** | 0.74 (± 0.27) |
| Sensitivity | 0.70 (± 0.07) | **0.72 (± 0.08)** | 0.68 (± 0.23) |
| $F_1$-Score | 0.70 (± 0.03) | **0.74 (± 0.06)** | 0.70 (± 0.16) |
| Accuracy | 0.69 (± 0.04) | **0.74 (± 0.07)** | 0.71 (± 0.13) |

been done for each of the five partitions. For comparison purpose, evaluation metrics using both DeCAF visual representation are listed. Tables 3 and 4 show the classification rates obtained on the k-fold cross validation sets. The evaluation rates achieved on the independent test set with $n$ = 21 participants are listed in Tables 5 and 6. The random split was done class-wise, to confirm the occurrence of both classes in the k-fold cross validation sets.

**Table 5. Prediction accuracy on HNR Study test set.** Prediction performance of the LSTM classification model using various image input types. The highlighted values are the best per evaluation metric. Evaluation was calculated on the independent test set with $n$ = 21 participants of the sub-study from the HNR Study. Visual representation were extracted using the ImageNet database [64].

|  | Original | Branded | Wide and Deep |
|---|---|---|---|
| Specificity | 0.82 | **0.91** | 0.64 |
| Sensitivity | 0.70 | **0.90** | **0.90** |
| $F_1$-Score | 0.74 | **0.90** | 0.78 |
| Accuracy | 0.76 | **0.90** | 0.76 |

**Table 6. Prediction accuracy on HNR Study test set.** Prediction performance of the LSTM classification model using various image input types. The highlighted values are the best per evaluation metric. Evaluation was calculated on the independent test set with $n$ = 21 participants of the sub-study from the HNR Study. Visual representation were extracted using the ChestX-Ray8 database [16].

|  | Original | Branded | Wide and Deep |
|---|---|---|---|
| Specificity | 0.73 | **0.91** | 0.64 |
| Sensitivity | 0.90 | **0.80** | 1 |
| $F_1$-Score | 0.82 | **0.84** | 0.83 |
| Accuracy | 0.81 | **0.86** | 0.81 |

**Table 7. Cross-validation prediction on ADNI Phase 1 dataset.** Prediction performance of the LSTM classification model using various image input types. The highlighted values are the best per evaluation metric. Evaluation was calculated on the $k = 5$-fold cross validation sets from the training set with $n = 561$ participants of the ADNI Phase 1 dataset. The values are the average and standard deviation rates across all $k = 5$-fold cross validation sets. Visual representation were extracted using the ImageNet database [64].

|  | Original | Branded | Wide and Deep |
|---|---|---|---|
| Specificity | 0.44 (± 0.08) | **0.54 (± 0.11)** | 0.47 (± 0.09) |
| Sensitivity | 0.82 (± 0.06) | **0.83 (± 0.12)** | 0.81 (± 0.03) |
| $F_1$-Score | 0.79 (± 0.04) | **0.81 (± 0.07)** | 0.80 (± 0.03)) |
| Accuracy | 0.69 (± 0.06) | **0.74 (± 0.09)** | 0.71 (± 0.04) |

**Table 8. Cross-validation prediction on ADNI Phase 1 dataset.** Prediction performance of the LSTM classification model using various image input types. The highlighted values are the best per evaluation metric. Evaluation was calculated on the $k = 5$-fold cross validation sets from the training set with $n = 561$ participants of the ADNI Phase 1 dataset. The values are the average and standard deviation rates across all $k = 5$-fold cross validation sets. Visual representation were extracted using the ChestX-Ray8 database [16].

|  | Original | Branded | Wide and Deep |
|---|---|---|---|
| Specificity | 0.41 (± 0.08) | **0.57 (± 0.09)** | 0.39 (± 0.05) |
| Sensitivity | 0.67 (± 0.04) | **0.71 (± 0.09)** | 0.70 (± 0.03) |
| $F_1$-Score | 0.67 (± 0.02) | **0.72 (± 0.06)** | 0.68 (± 0.00)) |
| Accuracy | 0.58 (± 0.03) | **0.64 (± 0.09)** | 0.59 (± 0.01) |

For the ADNI datatset, a $k = 5$-fold cross validation [72] was achieved by splitting the training set with 561 participants into 5 different partitions. From this, one partition is used as the validation set (10%) and the remaining 4 partitions (90%) are used for training. This has been done for each of the five partitions. Tables 7 and 8 show the classification rates obtained on the k-fold cross validation sets. The evaluation rates achieved on the independent test set with $n = 63$ participants are listed in Tables 9 and 10. The random split was done class-wise, to confirm the occurrence of both classes in the k-fold cross validation sets.

Fig 6 displays the Gradient-weighted Class Activation Mapping (Grad-CAM) [73] of the adopted LSTM model. The Grad-CAM shows visual explanations of the decisions made by the LSTM models, highlighting the important regions of the MRI scans used to distinguish between aMCI and controls. An ablation study was conducted prior to branding by omitting each clinical data variable. This ablation study gives insight regarding the information gain by applying sociodemographic data and APOE-$\varepsilon$4, which is listed in Tables 11 and 12 for the HNR Study dataset and in Tables 13 and 14 for the ADNI Phase 1 dataset.

**Table 9. Prediction accuracy on ADNI Phase 1 test set.** Prediction performance of the LSTM classification model using various image input types. The highlighted values are the best per evaluation metric. Evaluation was calculated on the independent test set with $n = 63$ participants of the ADNI Phase 1 dataset. Visual representation were extracted using the ImageNet database [64].

|  | Original | Branded | Wide and Deep |
|---|---|---|---|
| Specificity | 0.48 | **0.65** | 0.52 |
| Sensitivity | **0.85** | **0.85** | 0.77 |
| $F_1$-Score | 0.78 | **0.83** | 0.79 |
| Accuracy | 0.66 | **0.77** | 0.71 |

**Table 10. Prediction accuracy on ADNI Phase 1 test set.** Prediction performance of the LSTM classification model using various image input types. The highlighted values are the best per evaluation metric. Evaluation was calculated on the independent test set with $n$ = 63 participants of the ADNI Phase 1 dataset. Visual representation were extracted using the ChestX-Ray8 database [16].

|  | Original | Branded | Wide and Deep |
|---|---|---|---|
| Specificity | 0.43 | **0.57** | 0.48 |
| Sensitivity | **0.59** | **0.77** | 0.70 |
| $F_1$-Score | 0.61 | **0.72** | 0.69 |
| Accuracy | 0.53 | **0.76** | 0.61 |

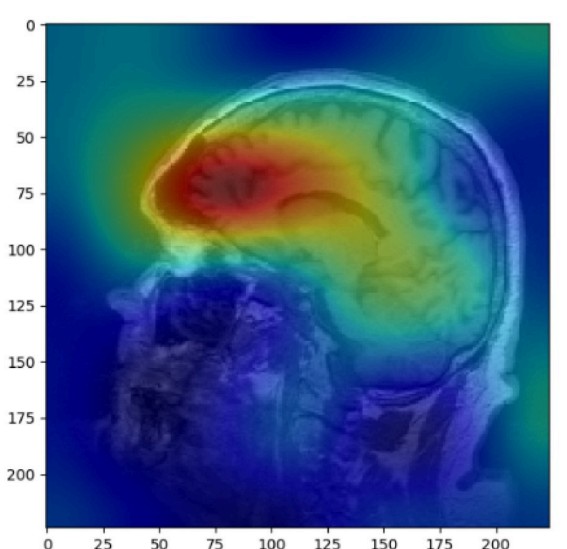 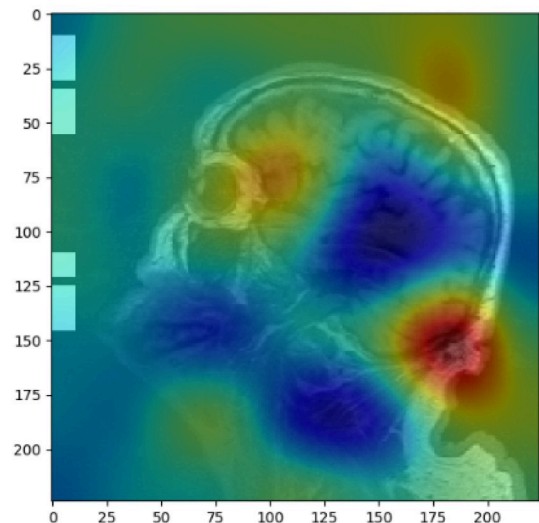

**Fig 6. Classification activation mapping.** Gradient-weighted Class Activation Mapping (Grad-CAM) image, highlighting important image regions used for distinguishing between aMCI and controls by the classification models. The 2D slice was randomly chosen from the sub-study of the HNR Study.

## Discussion

The proposed branding technique to obtain fused image representations of MRI scans with the sociodemographic data age, gender, education and marital status and APOE-$\varepsilon$4 genotype outperformed other inputs in all evaluation metrics in the independent test set. This could be shown for both the HNR Study and ADNI Phase 1 datasets.

**Table 11. Ablation study on HNR Study test set.** Prediction performance of the LSTM classification model on the ablation study. Each sociodemographic data variable, as well as the genetic data APOE-$\varepsilon$4 was subsequently omitted, prior to the MRI branding. Evaluation was calculated on the independent test set with $n$ = 21 participants of the sub-study from the HNR Study. Visual representation were extracted using the ImageNet database [64].

|  | Specificity | Sensitivity | $F_1$-Score | Accuracy |
|---|---|---|---|---|
| All data variables | **0.91** | **0.90** | **0.90** | **0.90** |
| Without age | 0.82 | **0.90** | 0.84 | 0.86 |
| Without APOE-$\varepsilon$4 | 0.91 | 0.70 | 0.78 | 0.81 |
| Without gender | **0.91** | 0.80 | 0.86 | 0.86 |
| Without education | 0.82 | 0.80 | 0.80 | 0.81 |
| Without marital status | **0.91** | 0.80 | 0.84 | 0.86 |

**Table 12. Ablation study on HNR Study test set.** Prediction performance of the LSTM classification model on the ablation study. Each sociodemographic data variable, as well as the genetic data APOE-$\varepsilon4$ was subsequently omitted, prior to the MRI branding. Evaluation was calculated on the independent test set with $n = 21$ participants of the sub-study from the HNR Study. Visual representation were extracted using the ChestX-Ray8 database [16].

|  | Specificity | Sensitivity | $F_1$-Score | Accuracy |
|---|---|---|---|---|
| All data variables | **0.91** | **0.80** | **0.84** | **0.86** |
| Without age | 0.74 | **0.80** | 0.78 | 0.76 |
| Without APOE-$\varepsilon4$ | 0.82 | 0.80 | 0.86 | 0.86 |
| Without gender | **0.73** | 0.70 | 0.82 | 0.81 |
| Without education | 0.74 | 0.70 | 0.78 | 0.76 |
| Without marital status | **0.77** | 0.80 | 0.78 | 0.76 |

**Table 13. Ablation study on ADNI Phase 1 test set.** Prediction performance of the LSTM classification model on the ablation study. Each sociodemographic data variable, as well as the genetic data APOE-$\varepsilon4$ was subsequently omitted, prior to the MRI branding. Evaluation was calculated on the independent test set with $n = 63$ participants of the ADNI Phase 1 dataset. Visual representation were extracted using the ImageNet database [64].

|  | Specificity | Sensitivity | $F_1$-Score | Accuracy |
|---|---|---|---|---|
| All data variables | **0.65** | **0.85** | **0.83** | **0.77** |
| Without age | 0.57 | 0.72 | 0.73 | 0.63 |
| Without APOE-$\varepsilon4$ | 0.39 | 0.80 | 0.74 | 0.65 |
| Without gender | 0.61 | 0.72 | 0.74 | 0.68 |
| Without education | 0.52 | 0.72 | 0.73 | 0.65 |
| Without marital status | 0.57 | 0.75 | 0.75 | 0.68 |

**Table 14. Ablation study on ADNI Phase 1 test set.** Prediction performance of the LSTM classification model on the ablation study. Each sociodemographic data variable, as well as the genetic data APOE-$\varepsilon4$ was subsequently omitted, prior to the MRI branding. Evaluation was calculated on the independent test set with $n = 63$ participants of the ADNI Phase 1 dataset. Visual representation were extracted using the ChestX-Ray8 database [16].

|  | Specificity | Sensitivity | $F_1$-Score | Accuracy |
|---|---|---|---|---|
| All data variables | **0.57** | **0.77** | **0.72** | **0.76** |
| Without age | 0.50 | 0.66 | 0.71 | 0.60 |
| Without APOE-$\varepsilon4$ | 0.58 | 0.69 | 0.68 | 0.62 |
| Without gender | 0.52 | 0.70 | 0.66 | 0.61 |
| Without education | 0.49 | 0.71 | 0.70 | 0.64 |
| Without marital status | 0.54 | 0.72 | 0.72 | 0.65 |

For the k-fold cross validation samples with $n = 99$ participants on the HNR Study dataset, the Wide and Deep input method achieved a higher sensitivity rate. However, for the specificity, precision and overall accuracy rate, the proposed method obtains better scores. The original image as input obtains better specificity, precision and accuracy rates on the test set than the Wide and Deep input method. Analyzing the ablation study with DeCAF representation extracted from Inception-v3 [39], the following findings can be taken:

- gender does not affect the overall specificity

- education has the greatest impact on all four evaluation values

- APOE-$\varepsilon4$ does not affect specificity

- marital status does not affect specificity

- age does not affect sensitivity

- Removing the genetic variable APOE-$\varepsilon$4 leeds to the highest decrease in the $F_1$-Score

- All applied sociodemographic data and APOE-$\varepsilon$4 have an impact on the overall $F_1$-Score

Analyzing the ablation study with DeCAF representation extracted with ChestX-Ray8 [16], the following findings can be taken:

- Age, education and marital status do not affect the overall sensitivity

- Removing age and education leads to the highest decrease in specificity

- Removing age and education leads to the highest decrease in the overall accuracy

- Removing age leeds to the highest decrease in the $F_1$-Score

- All applied sociodemographic data and APOE-$\varepsilon$4 have an impact on the overall $F_1$-Score

For the k-fold cross validation samples with $n$ = 561 participants on the ADNI Phase 1 dataset, the original input method achieved the same sensitivity rate. However, for the specificity, precision and overall accuracy rate, the proposed method obtains better scores. The original image as input obtains better specificity, precision and accuracy rates on the test set than the Wide and Deep input method. Analyzing the ablation study with DeCAF representation extracted with Inception-v3 [39], the following findings can be taken:

- gender does not affect the overall specificity

- education has a great impact on all four evaluation values

- Removing APOE-$\varepsilon$4 has the highest decrease on specificity

- Removing age has the highest decrease on the accuracy rate

- All applied sociodemographic data and APOE-$\varepsilon$4 have an impact on all four evaluation metrics

Analyzing the ablation study with DeCAF representation extracted with ChestX-Ray8 [16], the following findings can be taken:

- Removing education led to the highest decrease in specificity

- Age and education have a great impact on all four evaluation values

- Removing age has the highest decrease on sensitivity and accuracy rate

- Removing gender has the highest decrease on the $F_1$-Score

- All applied sociodemographic data and APOE-$\varepsilon$4 have an overall impact on all four evaluation metrics

As mentioned earlier, adequate fusion of selected features leads to enriched and consolidated visual representations. We show that combining several data input sources from the medical domain, proves to be a possible way for tackling challenging medical tasks.

Deep convolutional neural networks incorporate the ability to extract color information from RGB-images. MRI scans offer important insight into visual representation which can be applied for automatic structuring, such as classification, semantic tagging, and disease detection. However, they are only gray-scaled and thereby use the same color information redundantly for all 3 color channels. In the notion of fusing information to achieve medical image understanding, the MRI scans are enhanced after branding and prior to training the

classification models. The evaluation results show that augmenting contrast intensified and blurred image adaptions as two extra layers increases the model performance regarding classification and annotation between aMCI and controls.

It has to be kept in mind that we did not have any biomarker information that is specific for hallmark AD proteinopathies like amyloid beta deposition or phosphorylated tau. Thus, we cannot identify the underlying pathology in our aMCI cases.

In contrast to the ADNI data set the data in the HNR stem from a local cohort of German nationality in three neighborhood cities in the Ruhr Area. As a consequence this study population is rather homogenous both in cultural as in ethnic aspects. The HNR research group has consistently experienced similar obervations also in other fields, like CVD prediction.

Due to the limited number of participants in the applied datasets, there are limitations to the usage of standard end-to-end deep learning classification architecture. However to utilize the benefits of deep learning systems and examine its capabilities, DeCAF are adopted for visual representation. The evaluation metric rates on the independent test set show that taking advantage of large trained deep learning models such as ImageNet as feature representation, the aMCI vs control classification models are fed sufficient information and are capable of predicting clinical outcome.

Each 2D slice of an MRI contains information and dependencies about predecessor and successor slice. LSTM models have the ability to accumulate information, thus feeding every 2nd slice of the MRI scans was not only time efficient but led to positive results. By adopting an LSTM model over a 3D convolutional neural network, the computational time is reduced, as convolutional operations for the 2D convolutional layers are done across the x and y dimension only. We could show that LSTM models are capable of classifying between aMCI and controls using sociodemographic data and APOE-$\varepsilon$4, and deep convolutional activation features. The Grad-CAM results showing the visual explanations of the applied LSTM model are on first sight reasonable.

The presented approach can be applied to create computer-aided diagnosis systems for aMCI vs cognitively unimpaired, as well as semantic structuring and tagging systems in practical clinical situations. Radiologists and neurologists can use the classifier output as 'second opinion' in addition to peer discussions. Another application is to integrate the classifier output for a built-in preselection filter after MRI scans are taken. Suspected aMCI cases can be highlighted with this filter, hence reducing the number of images radiologists have to examine and indicating when to comprehensively screen. As structured and annotated data is fundamental for effective Information Retrieval (IR) systems, the proposed method can be integrated for the modeling and creation of IR systems. The classifier outputs are then adopted for prior content tagging. Such IR systems can be used by early medical practioners to filter aMCI vs cognitively impaired for learning purposes.

The findings of this proposed work require further evaluation on different functional neuroimaging techniques. For tackling the challenging medical task of early and preclinical detection of AD dementia, the fusion of various clinical data can be intensively experimented, as there are numerous input sources in the medical domain.

## Conclusion

This work presents an approach to combine sociodemographic data and APOE-$\varepsilon$4 with 1.5T MRI scans to create optimized classification models to distinguish between aMCI and controls. The fusion method enables an enriched image representation, as classification systems with multi-modal image features have proven to obtain higher prediction accuracies. Information fusion is obtained by encoding the values of the APOE-$\varepsilon$4 and sociodemographic data

variables: gender, marital status, age and education as markers, and branding these on the MRI scans, prior to training and prediction.

Two extra color layers denoting the contrast intensified and blurred image adaptions are augmented to simulate RGB-channeled images, which aims to use the characteristic of deep convolutional neural networks for color extraction as features for training. LSTM based RNNs are modeled as aMCI vs control classification models, as each 2D slice of a MRI scan contains dependencies between predecessor and successor slices. The output of the classification models are justified with visual explanations, denoting the important image regions used for decision making.

This works shows that fusing sociodemographic and genetic data from participants in a sub-study from the HNR Study and the ADNI Phase 1 datasets with MRI scans obtains enriched visual information that provides adequate representations, which is essential for creating effective automatic structuring systems, such as classification models, disease detection and semantic tagging. This is observed for both visual feature input techniques: DeCAF representations from 'Branded' images and 'Wide and Deep' image representation method.

Prospective modeling and evaluation of mild cognitive impairment classification systems can be based on different multi-modal image representation, as positive results have been presented in recent approaches and in this work. The proposed work pursues the way of several fusion techniques of features from different heterogeneous modalities in the medical domain for computer-aided diagnosis applications and can be adapted o 3D deep learning approaches by branding volume markers.

## Acknowledgments

The authors express their gratitude to all study participants of the HNR Study, the personnel of the HNR study center and the EBT-scanner facilities, the investigative group and all former employees of the HNR study. The authors also thank the Advisory Board of the HNR Study: T. Meinertz, Hamburg, Germany (Chair); C. Bode, Freiburg, Germany; P.J. de Feyter, Rotterdam, Netherlands; B. Güntert, Hall i.T., Austria; F. Gutzwiller, Bern, Switzerland; H. Heinen, Bonn, Germany; O. Hess (†), Bern, Switzerland; B. Klein (†), Essen, Germany; H. Löwel, Neuherberg, Germany; M. Reiser, Munich, Germany; G. Schmidt (†), Essen, Germany; M. Schwaiger, Munich, Germany; C. Steinmüller, Bonn, Germany; T. Theorell, Stockholm, Sweden; and S.N Willich, Berlin, Germany.

In this work we employed the database of the Alzheimer's Disease Neuroimaging Initiative (ADNI). Data collection and sharing for this project was funded by the Alzheimer's Disease Neuroimaging Initiative (ADNI) (National Institutes of Health Grant U01 AG024904) and DOD ADNI (Department of Defense award number W81XWH-12-2-0012). ADNI is funded by the National Institute on Aging, the National Institute of Biomedical Imaging and Bioengineering, and through generous contributions from the following: AbbVie, Alzheimer's Association; Alzheimer's Drug Discovery Foundation; Araclon Biotech; BioClinica, Inc.; Biogen; Bristol-Myers Squibb Company; CereSpir, Inc.; Cogstate; Eisai Inc.; Elan Pharmaceuticals, Inc.; Eli Lilly and Company; EuroImmun; F. Hoffmann-La Roche Ltd and its affiliated company Genentech, Inc.; Fujirebio; GE Healthcare; IXICO Ltd.; Janssen Alzheimer Immunotherapy Research & Development, LLC.; Johnson & Johnson Pharmaceutical Research & Development LLC.; Lumosity; Lundbeck; Merck & Co., Inc.; Meso Scale Diagnostics, LLC.; NeuroRx Research; Neurotrack Technologies; Novartis Pharmaceuticals Corporation; Pfizer Inc.; Piramal Imaging; Servier; Takeda Pharmaceutical Company; and Transition Therapeutics. The Canadian Institutes of Health Research is providing funds to support ADNI clinical sites in Canada. Private sector contributions are facilitated by the Foundation for the National

Institutes of Health (www.fnih.org). The grantee organization is the Northern California Institute for Research and Education, and the study is coordinated by the Alzheimer's Therapeutic Research Institute at the University of Southern California. ADNI data are disseminated by the Laboratory for Neuro Imaging at the University of Southern California.

Data used in preparation of this article were obtained from the Alzheimer's Disease Neuroimaging Initiative (ADNI) database (adni.loni.usc.edu). As such, the investigators within the ADNI contributed to the design and implementation of ADNI and/or provided data but did not participate in analysis or writing of this report. A complete listing of ADNI investigators can be found at: http://adni.loni.usc.edu/wp-content/uploads/how_to_apply/ADNI_Acknowledgement_List.pdf.

## Author Contributions

**Conceptualization:** Christoph M. Friedrich.

**Data curation:** Karl-Heinz Jöckel, Sara Schramm, Sarah Sanchez Hoffmann, Angela Winkler, Christian Weimar, Martha Jokisch.

**Funding acquisition:** Christoph M. Friedrich.

**Methodology:** Obioma Pelka, Christoph M. Friedrich.

**Resources:** Felix Nensa, Christoph Mönninghoff, Karl-Heinz Jöckel, Sara Schramm, Sarah Sanchez Hoffmann, Angela Winkler, Christian Weimar.

**Software:** Obioma Pelka.

**Supervision:** Christoph M. Friedrich, Felix Nensa.

**Validation:** Obioma Pelka, Felix Nensa, Christian Weimar, Martha Jokisch.

**Visualization:** Obioma Pelka.

**Writing – original draft:** Obioma Pelka, Louise Bloch, Martha Jokisch.

**Writing – review & editing:** Christoph M. Friedrich, Felix Nensa, Karl-Heinz Jöckel, Sara Schramm, Sarah Sanchez Hoffmann, Christian Weimar.

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
