## [Decision Letter · Decision Letter 0]

6 Jan 2020

PONE-D-19-27720

Sociodemographic and Genetic Data Augmentation for MRI-based Detection of Amnestic Mild Cognitive Impairment Using Deep Learning Methods

PLOS ONE

Dear Prof. Dr. Friedrich,

Thank you for submitting your manuscript to PLOS ONE. After careful consideration, we feel that it has merit but does not fully meet PLOS ONE’s publication criteria as it currently stands. Therefore, we invite you to submit a revised version of the manuscript that addresses the points raised during the review process.

ACADEMIC EDITOR: 

Dear authors

The majority of commented raised from the reviewers can be answered properly.

The second reviewer raised an issue regarding the reproducibility of the findings by using 

a closed database. I can understand that it is always possible to open a database to the public

due to many issues from the funding schemes.

My opinion is to reproduce your findings by using ADNI database. This will further strengthen your study.

We would appreciate receiving your revised manuscript by Feb 20 2020 11:59PM. To enhance the reproducibility of your results, we recommend that if applicable you deposit your laboratory protocols in protocols.io, where a protocol can be assigned its own identifier (DOI) such that it can be cited independently in the future. For instructions see: http://journals.plos.org/plosone/s/submission-guidelines#loc-laboratory-protocols

We look forward to receiving your revised manuscript.

Kind regards,

Stavros I. Dimitriadis

Academic Editor

PLOS ONE

Additional Editor Comments:

Dear authors

I have received the comments from two reviewers.

My recommendation is major revision with the suggestion of replicating your findings in an open database

like ADNI using the same design of cohort (age,gender,group distribution).

If you decide to revise your manuscript, I will be glad to send it back to reviewers for further evaluation.

Journal Requirements:

2. Please provide additional details regarding participant consent. In the Methods section, please ensure that you have specified (1) whether consent was informed and (2) what type you obtained (for instance, written or verbal). If your study included minors, state whether you obtained consent from parents or guardians. If the need for consent was waived by the ethics committee, please include this information.

3. Thank you for including your ethics statement:

"Ethics Statement from IRB of University Hospital Essen, Essen, Germany available dated 2009-10-20 to Prof. Dr. C. Weimar, registration number: 06-3116"

a. Please amend your current ethics statement to confirm that your named institutional review board or ethics committee specifically approved this study.

5. Thank you for stating the following in the Financial Disclosure section:

"The work of Obioma Pelka was partially funded by a

PhD grant from University of Applied Sciences and Arts

Dortmund, Germany.

The authors thank the Heinz Nixdorf Foundation [Chairman: Martin Nixdorf; Past Chairman: Dr jur. Gerhard Schmidt (†)], for their generous support of this study. Parts of the study were also supported by the German Research Council (DFG) [DFG project: EI 969/2-3, ER 155/6-1;6-2, HO 3314/2-1;2-2;2-3;4-3, INST 58219/32-1, JO 170/8-1, KN 885/3-1, PE 2309/2-1, SI 236/8-1;9-1;10-1,], the German Ministry of Education and Science [BMBF project: 01EG0401, 01GI0856, 01GI0860, 01GS0820_WB2-C, 01ER1001D, 01GI0205], the Ministry of Innovation, Science, Research and Technology, North Rhine-Westphalia (MIWFT-NRW), the Else Kröner-Fresenius-Stiftung [project: 2015_A119] and the German Social Accident Insurance [DGUV project: FF-FP295]. Furthermore the study was supported by the Competence Network for HIV/AIDS, the deanship of the university hospital and IFORES of the university Duisburg-Essen, the European Union, the German Competence Network Heart Failure, Kulturstiftung Essen, the Protein Research Unit within Europe (PURE) and the following companies: Celgene GmbH München, Imatron/GE-Imatron, Janssen, Merck KG, Philips, ResMed Foundation, Roche Diagnostics, Sarstedt AG&Co, Siemens HealthCare Diagnostics, Volkswagen Foundation. This substudy conducted in the Department of Neurology of the university hospital Essen was additionally supported by the Dr. Werner-Jackstädt Stiftung and Janssen."

We note that you received funding from a commercial source:  Celgene GmbH München, Imatron/GE-Imatron, Janssen, Merck KG, Philips, ResMed Foundation, Roche Diagnostics, Sarstedt AG&Co, Siemens HealthCare Diagnostics and the Volkswagen Foundation.

Reviewers' comments:

Reviewer's Responses to Questions

**Comments to the Author**

1. Is the manuscript technically sound, and do the data support the conclusions?

Reviewer #1: Yes

Reviewer #2: No

2. Has the statistical analysis been performed appropriately and rigorously? 

Reviewer #1: Yes

Reviewer #2: Yes

3. Have the authors made all data underlying the findings in their manuscript fully available?

Reviewer #1: Yes

Reviewer #2: No

4. Is the manuscript presented in an intelligible fashion and written in standard English?

Reviewer #1: Yes

Reviewer #2: Yes

5. Review Comments to the Author

Reviewer #1: The authors conducted an observational case-control study including 120 participants (61 amnestic Mild Cognitive Impairment (aMCI) and 59 cognitively unimpaired) of the Heinz Nixdorf Recall (HNR) Study. The aim of the study was to fuse sociodemographic data (age, marital status, education and gender) and genetic data (presence of an APOE4 allele) with Magnetic Resonance Imaging (MRI) scans, and, by this approach enable enriched multi-modal features (which represented the MRI scan visually) creating and modeling classification systems capable of detecting aMCI. To fully utilize the potential of Deep Convolutional Neural Networks- two extra color layers denoting contrast intensified and blurred image adaptations- were virtually augmented to each MRI scan, completing the Red-Green-Blue color channels. The Deep convolutional activation features were extracted from the average pooling layer of the deep learning system Inception v3. These features from the fused MRI scans were used as visual representation for the Long Short-Term Memory based Recurrent Neural Network classification model. The dataset consisting of 120 participants was split into a training and test set, containing 99 participants (aMCI=51 and controls=48) and 21 participants (aMCI=10 and controls=11), respectively. The test set was independent and not used for training or parameter optimization. Experimental results showed the proposed approach achieved 90% accuracy and 0.90 F1-Score at classification of aMCI vs. cognitively unimpaired participants.

The theme is interesting.The subject is relevant. The design of the study is clear. There are some difficulties in Introduction section and in describing methods of the study.

I would encourage the authors to review (major revision) the manuscript since the topic is relevant.

Following, I made some recommendations to the authors, by section.

The title: is good.

Abstract section: is good.

Introduction section:

1 - Page 2, line 11: “The amnestic MCI (aMCI) subtype has a high probability of progressing to AD”. Indeed, AD neuropathology could be present and related to aMCI by the time aMCI is diagnosed. In this case, aMCI should correspond to first symptomatic presentation of AD. Therefore, I suggest including the term "dementia" to AD instead of only AD.

2- The authors should explain previous works and knowledge development as a history. I suggest they avoid citation like this one in page 2, line 20: “As shown in [6–9], multi-modal representation achieves higher prediction rates in medical and biomedical annotation tasks...”. Also, in page 2, line 38: “In [17, 18], automatic generated keywords are combined with x-rays to obtain optimized body region classification”. And, in page 2, line 44: “In [22, 23], late fusion methods were applied, where decision values from several classifiers are fused to make the final classification prediction”. The authors frequently cite previous works by the references. I suggest they cite the authors first, as a history of development of knowledge, and, then, they should include the citation number.

Methods section:

3 - Page 3, line 88: the authors explain that the current study is a sub-study derived from the Heinz Nixdorf Recall (HNR) Study. They relate that the verbal memory, orientation/praxis, information processing speed, executive functions and verbal abilities were covered by the study protocol. Was the aMCI diagnosis based on the results of the short or on more comprehensive neuropsychological assessment?

4- What was considered the gold standard method for diagnosing aMCI?

5-When was the current study conducted?

6- Considering that current study is a sub-study derived from the Heinz Nixdorf Recall (HNR) Study, what were the inclusion and exclusion criteria?

7- Page 3, line 107: the authors explained “Because the MCI due to AD criteria by Albert et al. [42] were not yet published when the sub-study started, the Winblad et al. [43] MCI criteria were used to diagnose aMCI. Therefore, the diagnosis of aMCI was equivalent to the diagnosis of MCI due to AD without biomarker information representing the core clinical criteria as proposed by Albert et al".

Considering the study propose an approach to diagnose, absence of the AD biomarkers must be considered a limitation of the study.

8- The authors included sociodemographic characteristics and APOE-ε4 genotyping as study covariates. Why didn´t they include clinical characteristics, especially cardiovascular risk factors, as covariates? I think clinical data should be disponible, since the current study derived from The HNR Study, a population-based prospective cohort study which evaluate cardiovascular risk factors. They are considered risk factors to AD.

9-Page5, line 135: “Table 1 shows the distribution of the sociodemographic data variables age, gender,

education, marital status and genetic data variable APOE-ε4 genotype (defined as

“clinical data”) for aMCI and cognitively unimpaired controls on the applied sub

sample”.

Why did the authors consider genetic data as clinical data? Please, explain it.

10- Page 5, line 142: “The presented work proposes an approach to fuse clinical data with MRI scans,

enabling enriched multi-modal image representation”.

Is it possible to consider sociodemographic characteristics and APOE-ε4 genotype as clinical data?

11- Page 5, line 142: “This is fundamental for image classification and retrieval purposes, and is not limited to computer-aided decision systems for clinical diagnoses, as positive results have been presented for 2D Images in Pelka et al. [21]. The method of encoding clinical data onto 2D slices of an MRI scan is denoted in this work as “Branded“.

Since current studies have used 3D image-processing, using MRI 2D slices does not constitute a study limitation? Please, explain it better.

12-Page 6: Table 2 needs a legend with explanations.

13- I think statistical analysis needs to be more detailed.

Result section:

It is good.

Tables and figures are good.

Discussion section:

14- Discussion is good, based on the results. However, it is important to emphasize the limitations, especially lack of clinical data and of biomarkers of AD.

Conclusion:

Conclusions are based on the results. However, in page 11, line 325, the authors conclude “Information fusion is obtained by encoding the values of the clinical data variables: gender, marital status, APOE-ε4, age and education as markers and branding these on the MRI scans, prior to training and prediction.

One more time, I advise that it is important to pay attention to “clinical data variables”.

Sincerely.

Reviewer #2: Major comments

This paper presents an interesting work on ‘branding’ sociodemographic data and ApoE information to MRI data in a deep learning framework (CNN + LSTM) for amnestic MCI prediction. The four different inputs tested are pretty thorough design to evaluate the proposed method. However, this study, in its current form, still needs much more work both in the methodology and the study design.

Overall, methodologically, I think using the transfer learning on 2D MRI slices, with some kind of augmentation to fit in the RGB channels, and using LSTM to model information across slices are much like very early deep learning works in this medical imaging, when people tried to fit the medical image into successful deep learning models demonstrated elsewhere. I think the field has moved beyond it. This also relates to my point below, where large dataset can make a difference in methodological choice.

And for the specific study design, the main concern is the low number of subjects, as the authors also realized. We can see the large variance present in Table 3. There are many external large public datasets (ADNI, OASIS, AIBL, …) to test the algorithm. And due to the restriction on the data of this study, future researchers are also not able to compare their results with the result present in this study.

Minor comments:

dCNN and DeCafs are not good abbreviations for those well-known concepts.

I have heard little usage of the term ‘prediction rate’, maybe change to more common terms such as prediction accuracy.

The related work section is a bit confusing and unrelated, such as multiple image and text fusion references, the hearing loss paper. Some of the more related references might include:

Spasov, Simeon, et al. "A parameter-efficient deep learning approach to predict conversion from mild cognitive impairment to Alzheimer's disease." Neuroimage 189 (2019): 276-287.

Yang, Jie, et al. "Characterizing Alzheimer's Disease with Image and Genetic Biomarkers using Supervised Topic Models." IEEE journal of biomedical and health informatics (2019).

Lee, Garam, et al. "Predicting Alzheimer’s disease progression using multi-modal deep learning approach." Scientific reports 9.1 (2019): 1952.

It’s true ApoE information represents the most well-known genetic factor of AD, but having genetic data in the title while only using ApoE information is still misleading, I was expecting more SNPs included in the model.

The left image in the Grad-CAM seems to pinpoint frontal lobe and makes sense, perhaps it’s better to visualize in 3D, see e.g.

Feng, Xinyang, et al. "Deep Learning on MRI Affirms the Prominence of the Hippocampal Formation in Alzheimer's Disease Classification." bioRxiv (2018): 456277.

Using only clinical data (input #4) is not a good comparison, as the input is too different from the distribution represented in ImageNet (arguably MRI dataset is also pretty different), maybe just use some simple classifiers, for example, we can see from Table 1, ApoE itself should be able to achieve sensitivity better than 0.5.

As the dataset is from a longitudinal study, it would be interesting to including some longitudinal element either in the feature side or diagnosis side.

6. PLOS authors have the option to publish the peer review history of their article (what does this mean?). If published, this will include your full peer review and any attached files.

Reviewer #1: No

Reviewer #2: No

---

## [Author Response · Author response to Decision Letter 0]

16 Mar 2020

The responses are better readable in the corresponding document.

Editor

Thank you for the comment. The authors have followed the styling guidelines and made the appropriate changes.

2. Please provide additional details regarding participant consent. In the Methods section, please ensure that you have specified (1) whether consent was informed and (2) what type you obtained (for instance, written or verbal). If your study included minors, state whether you obtained consent from parents or guardians. If the need for consent was waived by the ethics committee, please include this information.

All participants provided written informed consent.

3. "Ethics Statement from IRB of University Hospital Essen, Essen, Germany available dated 2009-10-20 to Prof. Dr. C. Weimar, registration number: 06-3116" a. Please amend your current ethics statement to confirm that your named institutional review board or ethics committee specifically approved this study.

Thank you. The ethics statement has been correctly amended.

4. We note that you have indicated that data from this study are available upon request. a) If there are ethical or legal restrictions on sharing a de-identified data set, please explain them in detail (e.g., data contain potentially identifying or sensitive patient information) and who has imposed them (e.g., an ethics committee). Please also provide contact information for a data access committee, ethics committee, or other institutional body to which data requests may be sent.

De-identified data from this study are available upon request to Department of Neurology, University Hospital Essen, Germany pending approval by the study steering committee. Due to data security reasons (i.e., data contain potentially participant identifying information), the HNR Study does not allow sharing data as public use file. Data requests can be addressed to: recall@uk-essen.de

5. We note that you received funding from a commercial source: Celgene GmbH München, Imatron/GE-Imatron, Janssen, Merck KG, Philips, ResMed Foundation, Roche Diagnostics, Sarstedt AG&Co, Siemens HealthCare Diagnostics and the Volkswagen Foundation. Please provide an amended Competing Interests Statement that explicitly states this commercial funder, along with any other relevant declarations relating to employment, consultancy, patents, products in development, marketed products, etc.

Competing intereset were amended.

6. Please ensure that you refer to Table 9 in your text as, if accepted, production will need this reference to link the reader to the Table.

Table 9 has been correctly referred to.

7. Thank you for including your ethics statement on the submission form:

"Ethics Statement for the use of the HNR study population from IRB of University Hospital Essen, Essen, Germany available dated 2009-10-23 and 2012-06-06 to Prof. Dr. C. Weimar, registration number: 06-3116. All participants provided informed consent.

Details about the Ethics statement of the ADNI study population can be found at: https://adni.loni.usc.edu"

The ethics statement has been amended and added to the Methods section of the manuscript.

8. Thank you for clarifying in your Response to Reviewers that: "All participants provided written informed consent."

a) To help ensure that the wording of your manuscript is suitable for publication, would you please also add this statement at the beginning of the Methods section of your manuscript file.

This statement has been added to “Study Population” of the Methods section of the manuscript.

10. Thank you for providing the following data availability statement:

"De-identified data from this study are available upon request to Dr. Martha Jokisch pending approval by the study steering committee. Due to data security reasons (i.e., data contain potentially participant identifying information), the HNR Study does not allow sharing data as public use file. Data requests can be addressed to: recall@uk-essen.de

Additional Data used in preparation of this article were obtained from the Alzheimer’s Disease

Neuroimaging Initiative (ADNI) database (https://adni.loni.usc.edu). Details about data access is detailed there."

Before we proceed, please address the following points:

a) We note de-identified data are available upon request to Dr. Jokisch. PLOS policy does not allow authors to be the sole point of contact for data access queries. Please provide a non-author point of contact for fielding de-identified data access queries from future researchers.

We have provided a non-author point for data access queries and added this information to journal editor comments point 4.

b) Please provide any relevant accession codes, data set names, etc a future researcher may need when requesting access to the additional data obtained from the Alzheimer’s Disease Neuroimaging Initiative (ADNI) database.

There are no relevant accession codes or data set names.

c) Please also confirm the authors had no special access privileges others would not have to the data obtained from the Alzheimer’s Disease Neuroimaging Initiative (ADNI) database.

The authors had no special access privileges others would not have to the data obtained from the Alzheimer’s Disease Neuroimaging Initiative (ADNI) database. 

Reviewer 1

Introduction section:

1. Page 2, line 11: “The amnestic MCI (aMCI) subtype has a high probability of progressing to AD”. Indeed, AD neuropathology could be present and related to aMCI by the time aMCI is diagnosed. In this case1, aMCI should correspond to first symptomatic presentation of AD. Therefore, I suggest including the term "dementia" to AD instead of only AD.

We have included the term “dementia” accordingly.

2. The authors should explain previous works and knowledge development as a history. I suggest they avoid citation like this one in page 2, line 20: “As shown in [6–9], multi-modal representation achieves higher prediction rates in medical and biomedical annotation tasks...”. Also, in page 2, line 38: “In [17, 18], automatic generated keywords are combined with x-rays to obtain optimized body region classification”. And, in page 2, line 44: “In [22, 23], late fusion methods were applied, where decision values from several classifiers are fused to make the final classification prediction”. The authors frequently cite previous works by the references. I suggest they cite the authors first, as a history of development of knowledge, and, then, they should include the citation number.

The reference style was changed. The authors were first cited by their name and then the citation number.

Methods section:

3. Page 3, line 88: the authors explain that the current study is a sub-study derived from the Heinz Nixdorf Recall (HNR) Study. They relate that the verbal memory, orientation/praxis, information processing speed, executive functions and verbal abilities were covered by the study protocol. Was the aMCI diagnosis based on the results of the short or on more comprehensive neuropsychological assessment?

The aMCI diagnosis was based on a neurological and neuropsychological evaluation. The neuropsychological evaluation was performed by an experienced neuropsychologist using the more comprehensive assessment (see point below).

4. What was considered the gold standard method for diagnosing aMCI?

The gold standard method consisted of a neurological and neuropsychological evaluation. The neurological evaluation was performed by a senior neurologist. The neuropsychological examination was performed by an experienced neuropsychologist. The final decision about aMCI diagnosis was ultimately made by consensus agreement between the examining neurologist and neuropsychologist taking into account the medical history related to cognitive functioning, duration of such symptoms, the history of other medical illnesses and current treatment for each participant. Participants with aMCI had to fulfill the following criteria: Cognitive impairment in the verbal memory domain; presence of subjective cognitive decline; normal functional abilities of daily living and no dementia diagnosis.

We have now added this information to the “Evaluation of cognitive status and aMCI diagnosis” section as follows:

“The final decision about aMCI diagnosis was ultimately made by consensus agreement between the examining neurologist and neuropsychologist taking into account the medical history related to cognitive functioning, duration of such symptoms, the history of other medical illnesses and current treatment for each participant.”

5. When was the current study conducted?

The data used for these analyses were assessed from 2006 to 2009.This information has been added to the “Study population” section.

6. Considering that current study is a sub-study derived from the Heinz Nixdorf Recall (HNR) Study, what were the inclusion and exclusion criteria?

Inclusion criteria: A short cognitive assessment was performed in the HNR study (see Wege et al. 2011, Neuroepidemiology). A random sample of participants (aged 50–80 years) with impaired screening results (two subtests below the age- and education adjusted mean) and age-appropriate screening results were invited to the sub-study.

Exclusion criteria: Participants with dementia, severe depression (ADAS depression subscale score >4), Parkinson disease, mental retardation, severe alcohol consumption (for women: >20 g/day; for men: >40 g/day), known brain cancer, severe problems with the German language (foreign persons) and severe sensory impairment leading to invalid cognitive testing were excluded.

We have added this information to the “Study population” section.

7. Page 3, line 107: the authors explained “Because the MCI due to AD criteria by Albert et al. [42] were not yet published when the sub-study started, the Winblad et al. [43] MCI criteria were used to diagnose aMCI. Therefore, the diagnosis of aMCI was equivalent to the diagnosis of MCI due to AD without biomarker information representing the core clinical criteria as proposed by Albert et al".

We totally agree with the reviewer and have now added this limitation to our “discussion” section as follows:

“It has to be kept in mind that we did not have any biomarker information that is specific for hallmark AD proteinopathies like amyloid beta deposition or phosphorylated tau. Thus, we cannot identify the underlying pathology in our aMCI cases.”

8. The authors included sociodemographic characteristics and APOE-ε4 genotyping as study covariates. Why didn´t they include clinical characteristics, especially cardiovascular risk factors, as covariates?

We totally agree with the reviewer, however other clinical characteristics were not added as they were not part of the proof of principle.

9. Page5, line 135: “Table 1 shows the distribution of the sociodemographic data variables age, gender, education, marital status and genetic data variable APOE-ε4 genotype (defined as “clinical data”) for aMCI and cognitively unimpaired controls on the applied sub sample”. Why did the authors consider genetic data as clinical data? Please, explain it.

Thank you for the comment. To avoid confusion, the authors have changed “clinical data” to “participant data”. Participant data is further split into sociodemographic (age, gender, education and marital status) and genetic data (APOE)

10. Page 5, line 142: “The presented work proposes an approach to fuse clinical data with MRI scans, enabling enriched multi-modal image representation”. Is it possible to consider sociodemographic characteristics and APOE-ε4 genotype as clinical data?

Thank you for the comment. The authors have removed the misleading sentences.

11. Page 5, line 142: “This is fundamental for image classification and retrieval purposes, and is not limited to computer-aided decision systems for clinical diagnoses, as positive results have been presented for 2D Images in Pelka et al. [21]. The method of encoding clinical data onto 2D slices of an MRI scan is denoted in this work as “Branded“. Since current studies have used 3D image-processing, using MRI 2D slices does not constitute a study limitation? Please, explain it better.

The limitations regarding MRI 2D slices vs 3D have been added.

12. Page 6: Table 2 needs a legend with explanations.

Legends have been added to Table 2.

13. I think statistical analysis needs to be more detailed.

Additional analysis has been written. This is also the case for the second dataset (ADNI Phase 1) used for further evaluation.

Discussion section:

14. Discussion is good, based on the results. However, it is important to emphasize the limitations, especially lack of clinical data and of biomarkers of AD.

Thank you for the comment. The authors have removed the misleading sentences.

Conclusion:

15. Page 11, line 325, the authors conclude “Information fusion is obtained by encoding the values of the clinical data variables: gender, marital status, APOE-ε4, age and education as markers and branding these on the MRI scans, prior to training and prediction. I advise that it is important to pay attention to “clinical data variables”.

Thank you for the comment. The authors have removed the misleading sentences. 

Reviewer 2

Major comments:

Overall, methodologically, I think using the transfer learning on 2D MRI slices, with some kind of augmentation to fit in the RGB channels, and using LSTM to model information across slices are much like very early deep learning works in this medical imaging, when people tried to fit the medical image into successful deep learning models demonstrated elsewhere. I think the field has moved beyond it. This also relates to my point below, where large dataset can make a difference in methodological choice. For the specific study design, the main concern is the low number of subjects, as the authors also realized. We can see the large variance present in Table 3. There are many external large public datasets (ADNI, OASIS, AIBL, …) to test the algorithm. And due to the restriction on the data of this study, future researchers are also not able to compare their results with the result present in this study.

Thank you for the comment and we totally agree that an additional and open-accessible state-of-the-art dataset should be included. Limitations regarding 2D vs 3D models are now mentioned in the manuscript. The authors have further evaluated the proposed approach on the ADNI Phase 1 dataset. Using the same participants data structure and branding process, we reproduce similar predication scores for the classification of controls vs aMCI. This manuscript was submitted prior to the ADNI Data and Publications Committee and received approval regarding Data User Agreement. In this manuscript, ADNI is acknowledged, the data gathering is described in the Methods section, and included to the named authors.

Minor comments:

1. dCNN and DeCafs are not good abbreviations for those well-known concepts.

The abbreviation dCNN and DeCafs have been removed. DeCafs was changed throughout the manuscript to DeCAF.

2. I have heard little usage of the term ‘prediction rate’, maybe change to more common terms such as prediction accuracy.

Prediction rate was changed to prediction accuracy.

3. The related work section is a bit confusing and unrelated, such as multiple image and text fusion references, the hearing loss paper. Some of the more related references might include:

Spasov, Simeon, et al. "A parameter-efficient deep learning approach to predict conversion from mild cognitive impairment to Alzheimer's disease." Neuroimage 189 (2019): 276-287.

Yang, Jie, et al. "Characterizing Alzheimer's Disease with Image and Genetic Biomarkers using Supervised Topic Models." IEEE journal of biomedical and health informatics (2019).

Lee, Garam, et al. "Predicting Alzheimer’s disease progression using multi-modal deep learning approach." Scientific reports 9.1 (2019): 1952.

Thank you for the comment and we totally agree. Misleading and unrelated references were removed. The proposed research (Spasov et al., Yang et al. Anf Garam et al.) have been added to the Related Work section.

4. It’s true ApoE information represents the most well-known genetic factor of AD, but having genetic data in the title while only using ApoE information is still misleading

The misleading title has been changed. We have replaced genetic with merely ApoE.

5. The left image in the Grad-CAM seems to pinpoint frontal lobe and makes sense, perhaps it’s better to visualize in 3D, see e.g. Feng, Xinyang, et al. "Deep Learning on MRI Affirms the Prominence of the Hippocampal Formation in Alzheimer's Disease Classification." bioRxiv (2018): 456277.

Thank you for the comment. The adopted approach is a 2D classification model and can unfortunately not be used to create similar Grad-CAM visualization as seen in Feng et al.

6. Using only clinical data (input #4) is not a good comparison

Thank you for the comment. The usage of merely clinical data has been completely removed from the manuscript. As Reviewer 1 mentioned, the term “clinical data” is misleading and was changed to “participants data”

---

## [Decision Letter · Decision Letter 1]

1 Jun 2020

PONE-D-19-27720R1

Sociodemographic Data and APOE-ε4 Augmentation for MRI-based Detection of Amnestic Mild Cognitive Impairment Using Deep Learning Systems

PLOS ONE

Dear Dr. Friedrich,

Thank you for submitting your manuscript to PLOS ONE. After careful consideration, we feel that it has merit but does not fully meet PLOS ONE’s publication criteria as it currently stands. Therefore, we invite you to submit a revised version of the manuscript that addresses the points raised during the review process.

During the final round of review, one of the reviewers suggested a few more corrections regarding

the references, the explanation of a specific deep learning model with transfer learning and a new experiment

using only one portion of the dataset.

I suggest you to respond to these comments and revised properly the draft.

We look forward to receiving your revised manuscript.

Kind regards,

Stavros I. Dimitriadis

Academic Editor

PLOS ONE

Additional Editor Comments (if provided):

Reviewers suggested a few more comments during the final round.

I suggest to read carefully the comments and answer them one by one

in your revised letter.

Reviewers' comments:

Reviewer's Responses to Questions

**Comments to the Author**

1. If the authors have adequately addressed your comments raised in a previous round of review and you feel that this manuscript is now acceptable for publication, you may indicate that here to bypass the “Comments to the Author” section, enter your conflict of interest statement in the “Confidential to Editor” section, and submit your "Accept" recommendation.

Reviewer #1: All comments have been addressed

Reviewer #3: (No Response)

2. Is the manuscript technically sound, and do the data support the conclusions?

Reviewer #1: Yes

Reviewer #3: Yes

3. Has the statistical analysis been performed appropriately and rigorously? 

Reviewer #1: Yes

Reviewer #3: Yes

4. Have the authors made all data underlying the findings in their manuscript fully available?

Reviewer #1: Yes

Reviewer #3: Yes

5. Is the manuscript presented in an intelligible fashion and written in standard English?

Reviewer #1: Yes

Reviewer #3: Yes

6. Review Comments to the Author

Reviewer #1: The authors conducted an observational case-control study including 120 participants (amnestic Mild Cognitive Impairment (aMCI) and cognitively unimpaired) of the HNR Study. The aim of the study was to fuse sociodemographic data and APOE4 allele with Magnetic Resonance Imaging scans, and, by this approach, enable enriched multi-modal features, creating and modeling classification systems capable of detecting aMCI. To fully utilize the potential of Deep Convolutional Neural Networks- two extra color layers denoting contrast intensified and blurred image adaptations- were virtually augmented to each MRI scan, completing the Red-Green-Blue color channels. The proposed approach was evaluated on a sub-study of 120 participants (aMCI and cognitively unimpaired) of the HNR Study with a baseline model accuracy of 76%. Further evaluation was conducted on the ADNI Phase 1 dataset with 624 participants (aMCI and cognitively unimpaired) with a baseline model accuracy of 66.27%. Experimental results showned the proposed approach achieves 90% accuracy and 0.90 F1-Score at classification of aMCI vs. cognitively unimpaired participants on the HNR Study dataset, and 77% accuracy and 0.83 F1-Score on the ADNI dataset.

The authors answered all the reviewers questions and made the suggested amendments which contributed to enhance the quality of the manuscript.

Sincerily,

Reviewer #3: 1. Can you expand on the number of patients that were removed from each of the exclusion criteria’s?

2. Can you expand on why the ADNI dataset had lower performance despite having more data to train on.

3. On line 31-33 you state “However, a major 31 concern in the medical domain is the lack of publicly available large image data sets 32 like ImageNet [16]. This is due to the fact that detailed annotation of medical images is 33 time-consuming, prone to errors and restricted by data protection rules” Minor point but can you just change it from lack of publicly available image data sets to insufficient number of datasets. They exist they’re just not enough for all contexts see.

Wang X, Peng Y, Lu L, Lu Z, Bagheri M, Summers RM. ChestX-ray8: Hospital-scale Chest X-ray Database and Benchmarks on Weakly-Supervised Classification and Localization of Common Thorax Diseases. IEEE CVPR 2017

Poldrack, R.A.; Barch, D.M.; Mitchell, J.; Wager, T.; Wagner, A.D.; Devlin, J.T.; Cumba, C.; Koyejo, O.; Milham, M. Toward open sharing of task-based fMRI data: the OpenfMRI project. Front. Neuroinf. 2013, 7, 12.

4. 89-90 “deep learning model, thereby exploring the potential of Transfer 89 Learning [15] from pre-trained ImageNet models” I would imagine using transfer learning from something from a medical domain like chestx-ray8 database would be better?

5. Can you expand on the performance impact of the augmented 2 color layers?

6. You use inception v3 as the basis of the transfer learning. Can you expand on why you chose that instead of a model pretrained for something in a medical context. As I mention in point 3 they exist.

7. How does the model perform if you train it only on the HNR study dataset and then test it on the ADNI phase 1 dataset

7. PLOS authors have the option to publish the peer review history of their article (what does this mean?). If published, this will include your full peer review and any attached files.

Reviewer #1: No

Reviewer #3: Yes: Ross O'Hagan

---

## [Author Response · Author response to Decision Letter 1]

15 Jul 2020

Reviewer 3

1. Can you expand on the number of patients that were removed from each of the exclusion criteria’s?

Participants with dementia (n=7), severe depression (ADAS depression subscale score >4, n=13), Parkinson disease (n=5), mental retardation (n=2), severe alcohol consumption (for women: >20 g/day; for men: >40 g/day, n=2), known brain cancer (n=1), severe problems with the German language (foreign persons, n=9) and severe sensory impairment (n=2) leading to invalid cognitive testing were excluded from the sub-study.

2. Can you expand on why the ADNI dataset had lower performance despite having more data to train on.

In contrast to the ADNI data set the data in the HNR stem from a local cohort of German nationality in three neighborhood cities in the Ruhr Area. As a consequence this study population is rather homogenous both in cultural as in ethnic aspects. The HNR research group has consistently experienced similar obervations also in other fields, like CVD prediction.

3. On line 31-33 you state “However, a major concern in the medical domain is the lack of publicly available large image data sets like ImageNet [16]. This is due to the fact that detailed annotation of medical images is time-consuming, prone to errors and restricted by data protection rules” Minor point but can you just change it from lack of publicly available image data sets to insufficient number of datasets. They exist they’re just not enough for all contexts see.

Thank you for the comment. We have edited the statement and the required references (Wang et. al and Poldrack et. al) were added.

4. 89-90 “deep learning model, thereby exploring the potential of Transfer 89 Learning [15] from pre-trained ImageNet models” I would imagine using transfer learning from something from a medical domain like chestx-ray8 database would be better?

We adopted the chestx-ray8 database weights instead of imagenet to extract visual representations prior to training. This step was implemented for both the HNR study dataset, as well as the ADNI phase I dataset. The obtained evaluation metric scores are listed alongside, for comparison purposes.

5. Can you expand on the performance impact of the augmented 2 color layers?

Thank you for the comment. We have added information regarding the prediction rate improvements.

6. You use inception v3 as the basis of the transfer learning. Can you expand on why you chose that instead of a model pretrained for something in a medical context. As I mention in point 3 they exist.

We have included the prediction performance using the classification model with transfer learning using the chestx-ray8 database.

7. How does the model perform if you train it only on the HNR study dataset and then test it on the ADNI phase 1 dataset

Thank you for the comment. This experiment was performed and obtained less favourable prediction rates.

---

## [Editor Report · Decision Letter 2]

16 Jul 2020

Sociodemographic Data and APOE-ε4 Augmentation for MRI-based Detection of Amnestic Mild Cognitive Impairment Using Deep Learning Systems

PONE-D-19-27720R2

Dear Dr. Friedrich,

We’re pleased to inform you that your manuscript has been judged scientifically suitable for publication and will be formally accepted for publication once it meets all outstanding technical requirements.

Kind regards,

Stavros I. Dimitriadis

Academic Editor

PLOS ONE

Additional Editor Comments (optional):

After carefully reading of your answers to reviewers' comments, I recommend the acceptance of your manuscript

in its current form.
---

## [Editor Report · Acceptance letter]

16 Sep 2020

PONE-D-19-27720R2

Sociodemographic Data and APOE-ε4 Augmentation for MRI-based Detection of Amnestic Mild Cognitive Impairment Using Deep Learning Systems

Dear Dr. Friedrich:

I'm pleased to inform you that your manuscript has been deemed suitable for publication in PLOS ONE. Congratulations! Your manuscript is now with our production department.

Kind regards,

on behalf of

Dr. Stavros I. Dimitriadis 

Academic Editor

PLOS ONE